# Ancient Jomon genome sequence analysis sheds light on migration patterns of early East Asian populations

Takashi Gakuhari [1,2,3,21], Shigeki Nakagome [4,21], Simon Rasmussen[5], Morten E. Allentoft[5,6], Takehiro Sato [7], Thorfinn Korneliussen[5], Blánaid Ní Chuinneagáin[4], Hiromi Matsumae [3], Kae Koganebuchi[3], Ryan Schmidt[3], Souichiro Mizushima[8], Osamu Kondo [9], Nobuo Shigehara[10], Minoru Yoneda[11], Ryosuke Kimura [12], Hajime Ishida [12], Tadayuki Masuyama[13], Yasuhiro Yamada [14], Atsushi Tajima [7], Hiroki Shibata[15], Atsushi Toyoda [16], Toshiyuki Tsurumoto[17], Tetsuaki Wakebe[17], Hiromi Shitara[18], Tsunehiko Hanihara[3], Eske Willerslev [5,19,20], Martin Sikora [5✉] & Hiroki Oota [3,9✉]

Anatomically modern humans reached East Asia more than 40,000 years ago. However, key questions still remain unanswered with regard to the route(s) and the number of wave(s) in the dispersal into East Eurasia. Ancient genomes at the edge of the region may elucidate a more detailed picture of the peopling of East Eurasia. Here, we analyze the whole-genome sequence of a 2,500-year-old individual (IK002) from the main-island of Japan that is characterized with a typical Jomon culture. The phylogenetic analyses support multiple waves of migration, with IK002 forming a basal lineage to the East and Northeast Asian genomes examined, likely representing some of the earliest-wave migrants who went north from Southeast Asia to East Asia. Furthermore, IK002 shows strong genetic affinity with the indigenous Taiwan aborigines, which may support a coastal route of the Jomon-ancestry migration. This study highlights the power of ancient genomics to provide new insights into the complex history of human migration into East Eurasia.

[1] Center for Cultural Resource Studies, College of Human and Social Sciences, Kanazawa University, Kanazawa, Japan. [2] Institute for Frontier Science Initiative, Kanazawa University, Kanazawa, Japan. [3] Kitasato University School of Medicine, Sagamihara, Japan. [4] School of Medicine, Trinity College Dublin, the University of Dublin, Dublin, Ireland. [5] Lundbeck Foundation GeoGenetics Centre, GLOBE Institute, University of Copenhagen, Copenhagen, Denmark. [6] Trace and Environmental DNA (TrEnD) laboratory, School of Molecular and Life Sciences, Curtin University, Perth, WA, Australia. [7] Department of Bioinformatics and Genomics, Graduate School of Advanced Preventive Medical Sciences, Kanazawa University, Kanazawa, Japan. [8] Department of Anatomy, St. Marianna University School of Medicine, Kawasaki, Japan. [9] Department of Biological Sciences, Graduate School of Science, The University of Tokyo, Tokyo, Japan. [10] Nara National Research Institute for Cultural Properties, Nara, Japan. [11] The University Museum, The University of Tokyo, Tokyo, Japan. [12] Graduate School of Medicine, University of the Ryukyus, Nishihara, Japan. [13] Educational Committee of Tahara City, Tahara, Japan. [14] National Museum of Japanese History, Sakura, Japan. [15] Division of Genomics, Medical Institute of Bioregulation, Kyushu University, Fukuoka, Japan. [16] National Institute of Genetics, Mishima, Japan. [17] Department of Macroscopic Anatomy, Nagasaki University Graduate School of Biomedical Science, Nagasaki, Japan. [18] Department of Archaeology, The University of Tokyo, Tokyo, Japan. [19] GeoGenetics Groups, Department of Zoology, University of Cambridge, Cambridge, UK. [20] Wellcome Trust Sanger Institute, Hinxton, UK. [21] These authors contributed equally: Takashi Gakuhari, Shigeki Nakagome. ✉email: martin.sikora@snm.ku.dk; hiroki_oota@bs.s.u-tokyo.ac.jp

After the major Out-of-Africa dispersal of *Homo sapiens* around 60,000 years ago (60 kya), modern humans rapidly expanded across the vast landscapes of Eurasia[1]. Both fossil and ancient genomic evidence suggest that groups ancestrally related to present-day East Asians were present in eastern China by as early as 40 kya[2]. Two major routes for these dispersals have been proposed, either from the northern or southern parts of the Himalaya mountains[1,3–6]. Population genomic studies on present-day humans[7,8] have exclusively supported the southern route origin of East Asian populations. On the other hand, the archaeological record provides strong support for the northern route as the origin of human activity, particularly for the arrival at the Japanese archipelago located at the east end of Eurasian continent. The oldest use of Upper Paleolithic stone tools goes back 38,000 years, and microblades, likely originated from an area around Lake Baikal in Central Siberia, are found in the northern island (i.e., Hokkaido; ~25 kya) and main-island (i.e., Honshu; ~20 kya) of the Japanese archipelago[9]. However, few human remains were found from the Upper Paleolithic sites in the archipelago. The Jomon culture started >16 kya, characterized by a hunter-fisher-gathering lifestyle with the earliest use of pottery in the world[10]. This Jomon culture lasted until a start of rice cultivation which brought by people migrated from the Eurasian continent, plausibly through the Korean peninsula, to northern parts of Kyushu island in the Japanese archipelago 3 kya. Several lines of archaeological evidence support the cultural continuity from the Upper-Paleolithic to the Jomon period, providing a hypothesis that the Jomon people are direct descendants of Upper-Paleolithic people who likely remained isolated in the archipelago until the end of Last Glacial Maximum[9,11,12]. Therefore, ancient genomics of the Jomon can provide new insights into the origin and migration history of East Asians.

A critical challenge for ancient genomics with samples from the Japanese archipelago is the inherent nature of warm and humid climate conditions except for the most north island, Hokkaido, and the soils indicating strong acidity because of the volcanic islands, which generally result in poor DNA preservation[13,14]. Though whole-genome sequences of two Hokkaido Jomon individuals dated to be 3500–3800-year-old were recently published with sufficient coverage[15], a partial genome of a 3000-year-old Jomon individual from the east-north part of Honshu Japan was reported, with very limited coverage (~0.03-fold) due to the poor preservation[16]. To identify the origin of the Jomon people, we sequenced the genome of a 2500-year-old Jomon individual (IK002) excavated from the central part of Honshu to 1.85-fold genomic coverage. Comparing this IK002 genome with ancient Southeast Asians, we previously reported genetic affinity between IK002 and the 8000 years old Hòabìnhian hunter-gatherer[17]. This direct evidence on the link between the Jomon and Southeast Asians, thus, suggests the southern route origin of the Jomon lineage. Nevertheless, key questions still remain as to (1) whether the Jomon were the direct descendant of the Upper Paleolithic people who were the first migrants into the Japanese archipelago and (2) whether the Jomon, as well as present-day East Asians, retain ancestral relationships with people who took the northern route.

Here, we test the deep divergence of the Jomon lineage and the impacts of southern- versus northern-route ancestry on the genetic makeup of the Jomon. The Jomon forms a lineage basal to both ancient and present-day East Asians; this deep origin supports the hypothesis that the Jomon were direct descendants of the Upper Paleolithic people. Furthermore, the Jomon has strong genetic affinities with the indigenous Taiwan aborigines. Our study shows that the Jomon-related ancestry is one of the earliest-

wave migrants who might have taken a coastal route on the way from Southeast Asia toward East Asia.

## Results

**Ancient DNA sequencing of Jomon specimens.** Initially, we extracted DNA of 12 individuals from three sites (Supplementary Fig. 1), which were accompanied by remains associated with the Jomon culture and had no evidence for influence from the following culture called Yayoi. The endogenous DNA contents for ten out of the twelve individuals were <1.0% due to poor DNA preservation, while those of two individuals, IK002 and HG02, were more than 1.0% (Supplementary Data 1). From those remaining two individuals, only the ~2500 years old IK002 exhibited typical patterns of DNA damage expected for ancient remains[18–20] (Supplementary Fig. 2), which was then selected for whole-genome sequencing. A total of 29 double-stranded sequencing libraries were generated using DNA extracted from the teeth and petrous bone of IK002, yielding endogenous DNA contents ranging from 1.14 to 19.09% (Supplementary Data 2). The libraries were sequenced to average coverages of 1.85-fold for autosomal genome and 146-fold for mitochondrial genome, with low estimated contamination rate of 0.5% (95% CI: 0.01–2.2%, Supplementary Fig. 3)[21]. We found IK002 to be assigned to mitochondrial haplogroup N9b1, which is rare among present-day Japanese people (<2.0%), but common in the Jomon[22,23]. We determined the genetic sex with the method based on the ratio of Y chromosome to X chromosome; the estimate of <0.3 supports that IK002 is female, which is consistent with morphological assessments.

**The origin of IK002.** To make inferences on the genetic relationship of IK002 with geographically diverse human populations, we merged the IK002 genomic data with a diverse panel of previously published ancient genomes[24–30], as well as 300 high-coverage present-day genomes from the Simons Genome Diversity Project[31]. We extracted genotypes for a set of 2,043,687 SNP sites included in the "2240K" capture panel[32]. First, we characterized IK002 in the context of worldwide populations using principal component analysis (PCA)[33,34]. We found that IK002 sat in between present-day East Asians and a cluster of ancient Hòabìnhian hunter-gatherers and the Upper-Paleolithic (40 kya) individual from Tiányuán Cave[17,30,35] (Fig. 1a). Second, the Honshu Jomon, IK002, closely clusters with the two Hokkaido Jomon, F23 and F5; we confirm that IK002 and the Hokkaido Jomon form a clade to the exclusion of other populations using $f_4$-statistics (Supplementary Data 3). Henceforth, we will use IK002 as the representative of the Jomon people in this paper. Third, when using a smaller number of SNPs (41,264 SNPs) including the present-day Ainu[36] from Hokkaido (Supplementary Fig. 1), IK002 clusters with the Hokkaido Ainu (Supplementary Fig. 4), supporting previous findings that the Hokkaido Ainu are direct descendants of the Jomon people[16,36–43]. Outgroup-$f_3$ statistics support those PCA clustering, with IK002 sharing most genetic drift with the Hokkaido Jomon, followed by the Ainu (Supplementary Fig. 5 & Supplementary Data 4). Thus, our results indicate that IK002 is genetically distinct from present-day people in East Eurasia or even in Japan, with the exception of the Hokkaido Ainu.

Subsequently, we carried out model-based unsupervised clustering using ADMIXTURE[44] (Supplementary Fig. 6). Assuming $K = 15$ ancestral clusters (Fig. 1b), an ancestral component unique to IK002 appears, which is the most prevalent in the Hokkaido Ainu (average 79.3%). This component is also shared with present-day Honshu Japanese as well as Ulchi (9.8% and

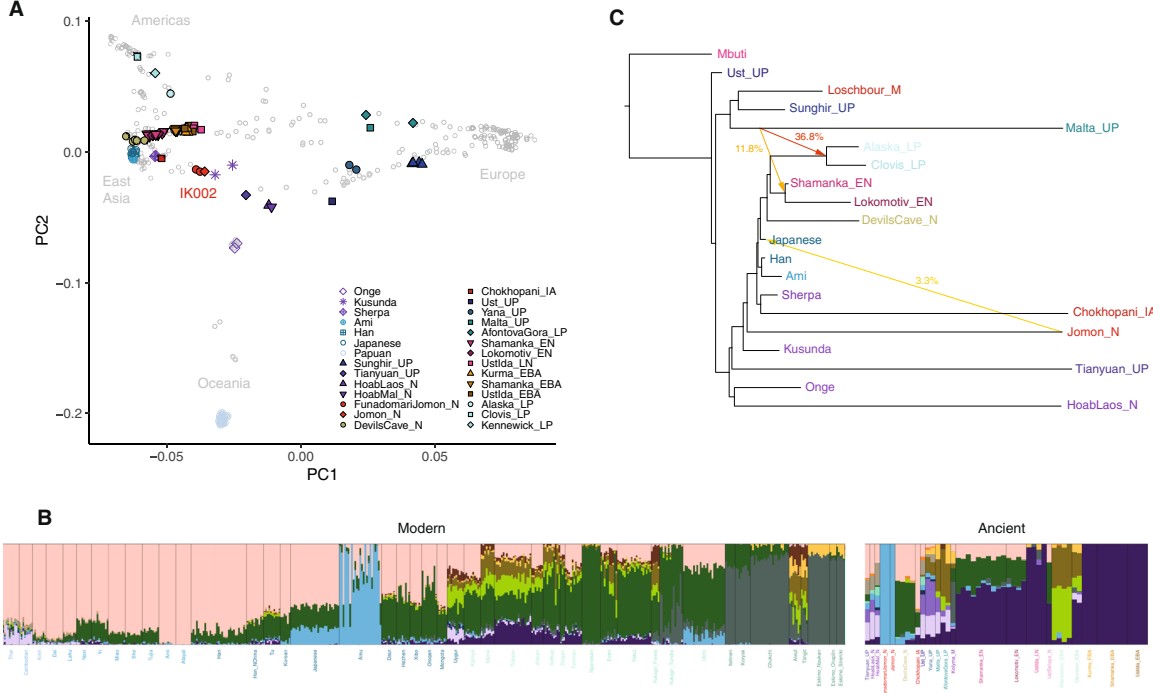

**Fig. 1 Genetic structure of present-day and ancient Eurasian and Ikawazu Jomon. a** Principal component analysis (PCA) of ancient and present-day individuals from worldwide populations after the out-of-Africa expansion. Gray labels represent population codes showing coordinates for individuals. Colored circles indicate ancient individuals. **b** ADMIXTURE ancestry components ($K = 15$) for ancient and selected contemporary individuals. The color of light blue represents the component of IK002, which is shared with the present-day Japanese and Ulchi. **c** Maximum-likelihood phylogenetic tree (*TreeMix*) with bootstrap support of 100% unless indicated otherwise. The tree shows phylogenetic relationship among present-day Southeast/East Asians, Northeast Siberians, Native Americans, and ancient East Eurasians. Mbuti are the present-day Africans; Ust'Ishim is an Upper-Paleolithic individual (45 kya) from Western Siberia[83]; Mal'ta (MA-1)[25] and Sunghir is Upper-Paleolithic individuals (24 kya and 34 kya)[29], and Loschbour is a Mesolithic individual from West Eurasia[88]; La368 is a pre-Neolithic Hòabìnhian hunter-gatherer (8.0 kya) in Laos and Önge is the present-day hunter-gatherers in the Andaman island, both of who are from Southeast Asia[17]; Tiányuán is an Upper-Paleolithic individual (40 kya) in Beijing, China[35]; Kusunda are the present-day minority people in Nepal; Chokhopani is an Iron-age individual (3.0–2.4 kya) and Sherpa are the present-day minority people, both of who are in Tibet[6]; Han, Ami and main-island Japanese are the present-day East Asians[31]; Devils Gate Cave is a Neolithic individual (8.0 kya) in the Primorye region of Northeast Siberia, and Shamanka and Lokomotive are Early-Neolithic individuals (8.0 kya) from Central Siberia, respectively[47]; USR1 and Clovis are late-Paleolithic individuals (11.5 kya and 12.6 kya) in Alaska and Montana, respectively[49,89]. Colored arrows represent the migration pathways and signals of admixture among all datasets. The migration weight represents the fraction of ancestry derived from the migration edge.

6.0%, respectively) (Fig. 1b). Those results also support the strong genetic affinity between IK002 and the Hokkaido Ainu.

We used ALDER[45] in order to date the timing of admixture in populations with Jomon ancestry. Using IK002 and the Hokkaido Jomon as a merged source population representing Jomon ancestry, and present-day Han Chinese as the second source representing mainland East Asian ancestry, we estimated the admixture in present-day Honshu Japanese to be between 60 and 77 generations ago (~1700–2200 years ago assuming 29 years/ generation), which is slightly earlier than previous estimates[8] but more consistent with the archaeological record (Supplementary Data 5). This indicates the admixture started and continuously occurred after the Yayoi period. For the Ulchi we estimated a more recent timing (31–47 generations ago) consistent with the higher variance in the IK002 component observed in ADMIX-TURE. Finally, we detected more recent (17–25 generations ago) admixture for the Hokkaido Ainu, likely a consequence of still ongoing gene flow between the Hokkaido Ainu and Honshu Japanese. The estimates of admixture timing are consistent when replacing Han with Korean, Ami or Devil's Gate cave as mainland East Asian source population, and exponential curves from a single admixture event fit the observed LD curve well (Supplementary Fig. 7 & Supplementary Data 5).

To further explore the deep relationships between the Jomon and other Eurasian populations, we used *TreeMix*[46] to reconstruct admixture graphs of IK002 and 18 ancient and present-day Eurasians and Native Americans (Fig. 1c & Supplementary Fig. 8). We found the IK002 lineage placed basal to the divergence between ancient and present-day Tibetans[6,31] and to the common ancestor of the remaining ancient/present-day East Eurasians[31,47,48] and Native Americans[49,50]. These genetic relationships are stable across different numbers of migration incorporated into the analysis. Major gene flow events recovered include the well-documented contribution of the Mal'ta individual (MA-1) to the ancestor of Native Americans[25,49], as well as a contribution of IK002 to present-day mainland Japanese ($m = 3$–8; Supplementary Fig. 8). IK002 can be modeled as a basal lineage to East Asians, Northeast Asians/East Siberians, and Native Americans, supporting a scenario in which their ancestors arrived through the southern route and migrated from Southeast Asia toward Northeast Asia[7,17]. However, regarding Native Americans, high genetic contributions (11.8–36.8%) were detected from the Upper Paleolithic individual, MA-1, which means that Native Americans were admixture between the southern and the northern routes as shown in Raghavan et al. (2014). The divergence of IK002 from the ancestors of continental East Asians therefore likely predates the split between East Asians and Native Americans, which has been previously estimated at 26 kya[49]. Thus, our *TreeMix* results support the hypothesis that IK002 is a direct descendant of the people who brought the Upper

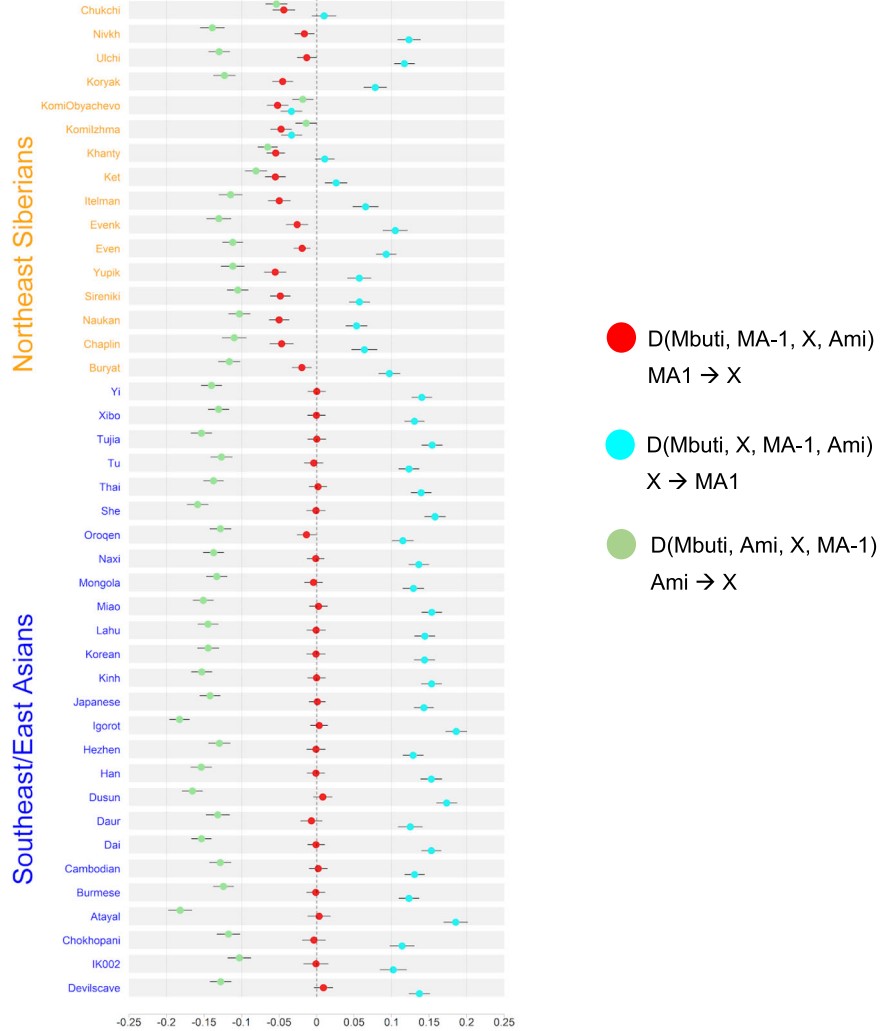

**Fig. 2 Exploring genetic affinities of IK002 within Northeast Siberians and Southeast/East Asians, respectively.** Three different $D$ values are plotted with different colors; $D$(Mbuti, MA-1; $X$, Ami) in red, $D$(Mbuti, $X$; MA-1, Ami) in cyan, and $D$(Mbuti, Ami; $X$, MA-1) in green. Error bars show three standard deviation, and the vertical dotted and dashed lines indicate $D = 0$ and $D = -0.2, -0.1, 0.1,$ and $0.2$.

Paleolithic stone tools 38,000 years ago into the Japanese archipelago.

**The impacts of the Northern route migration into East Asia.** Taking advantage of the earliest divergence of the IK002 (Fig. 1c & Supplementary Fig. 8), we address a question if the Upper-Paleolithic people who took the northern route of the Himalayas mountains to arrive east Eurasia made genetic contribution to populations migrated from Southeast Asia. Under the assumption that MA-1 is a descendant of a northern route wave, we tested gene flow from MA-1 to IK002, as well as to the other ancient and present-day Southeast/East Asians and Northeast Asians/East Siberians by three different forms of $D$ statistics: $D$(Mbuti, MA-1; $X$, Ami), $D$(Mbuti, $X$; MA-1, Ami), and $D$(Mbuti, Ami; $X$, MA-1).

The first $D$ statistics (shown as red in Fig. 2) provides results consistent with previous findings on the prevalence of MA-1 ancestry in the present-day Northeast Asians/East Siberians ($Z < -3$; $p < 0.003$, Supplementary Data 6)[25], while none of Southeast/East Asians, except for Oroqen, shows a significant deviation from zero. The tree relationships observed in Fig. 1c are confirmed from the other two different forms between Ami and all of the tested populations with some variation that is mostly explained by the

MA-1 gene flow (cyan and green in Fig. 2, Supplementary Data 7 & Supplementary Data 8). Therefore, there was no detectable signature of gene flow from MA-1 to the ancient/present-day Southeast/East Asians including IK002.

**Remnant of the Jomon-related ancestry in the coastal region.** The old divergence of IK002 (Fig. 1c) implies a negligible contribution to later ancient and present-day mainland East Asian groups. We further tested this implication by using $f_4$-statistics with the form of (Mbuti, IK002; $X$, Chokhopani). If IK002 was a true outgroup to later East Asian groups, this statistic is expected to be zero for any test population $X$. However, we find that together with Japanese, present-day Taiwan aborigines (i.e., Ami and Atayal), as well as populations from the Okhotsk-Primorye region (i.e., Ulchi and Nivhk), also showed a significant ($Z < -3$; $p < 0.003$) excess of allele sharing with IK002. Populations in the inland of the eastern part of the Eurasian continent on the other hand were consistent with forming a clade with Chokhopani (Fig. 3), which suggests the presence of Ikawazu Jomon-related ancestry in the present-day coastal populations in East Asia. The signal is also present in the Neolithic individuals from Devil's Gate Cave in the Primorye region ($Z < -3$; $p < 0.003$;

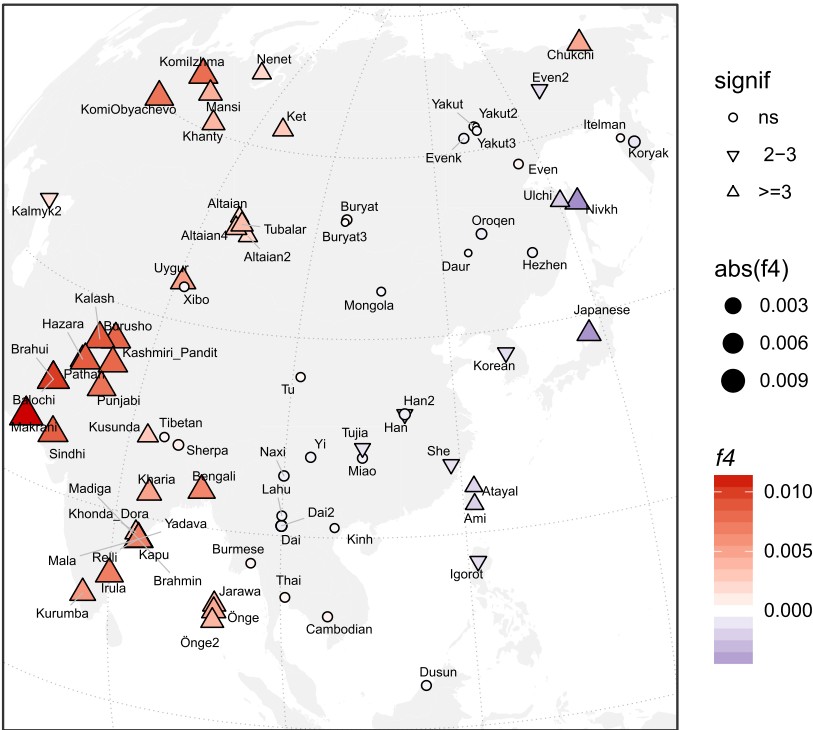

**Fig. 3 Heatmap of $f_4$-statistics comparing Eurasian populations to the Ikawazu Individual.** Heatmaps of $f_4$(Mbuti, IK002;$X$, Chokhopani), where $X$ are the present-day East Eurasian populations. The color and size represents the value of $f_4$-statistics. The shape represents statistical significances of genetic affinities based on $Z$-score. Triangle label means statistical significance with $|Z| > 3$ ($P < 0.01$), inverted triangle means weak significance with $|Z| = 2–3$ and circle means non-significance with $|Z| < 2$ ($P > 0.05$).

Supplementary Fig. 9), suggesting that, at 8 kya, populations who had IK002-related ancestry in the region had already been largely but not completely replaced by later migrations. Interestingly, the genetic affinity to IK002 was found only in the coastal region but not in the inland for both ancient and present-day populations (Fig. 3).

We carried out admixture graph modeling to further characterize the contributions of IK002-related ancestry to other East Asian populations. To that end, we first fit a backbone graph including ancient genomes representative of major divergences among East Asian lineages: IK002 (early dispersal); Chokhopani (later dispersal, East Asia), and Shamanka (later dispersal, Siberia). Test populations of interest were then modeled as three-way mixtures of early (IK002) and later (Chokhopani, Shamanka) dispersal lineages, using a grid search of admixture proportions within *qpGraph*. Consistent with the results from the $f_4$-statistics, we find that models without contribution from IK002 result in poor fit scores for Japanese, Devil's Gate Cave and Ami, as opposed to inland groups such as Han which do not require IK002-related ancestry (Supplementary Fig. 10). The range of admixture fractions with good model fit is generally quite wide, with best fit models showing IK002-related contributions of 8%, 4 and 41% into Japanese, Devil's Gate Cave and Ami, respectively (Supplementary Fig. 10 & Supplementary Fig. 11). While the substantial contribution into Ami seems at odds with the lower $f_4$-statistics compared with Japanese, the lineage admixing with Ami shares only a very short branch with IK002, suggesting a contribution from a distinct group with an early divergence from the IK002 lineage. We note that this backbone graph fitting assumed an unadmixed Jomon lineage, as opposed to a previously suggested dual-ancestry model where Jomon is admixed between Önge- and Ami-related ancestry. This alternative base model provides an equivalent admixture graph fit, however we find no evidence for shared genetic drift between the Önge and the

ancestral Jomon lineage in *qpGraph* (Supplementary Fig. 12a & Supplementary Fig. 12b), or using direct $f_4$-statistics (Supplementary Data 9 & Supplementary Data 10). Additional sampling of early East Asian human remains will be needed to further resolve the relationships among these deep lineages, but nevertheless either model supports the deep origins of Jomon.

## Discussion

This study takes advantage of whole-genome sequence data from the 2500 years old Jomon individual, IK002, dissecting the origins of present-day East Asians. IK002 is modeled as a basal lineage to East Asians, Northeast Asians/East Siberians, and Native Americans (basal East Eurasians, bEE) after the divergence between Tiányuán and the ancestor of hunters-gatherers in Southeast Asia (Fig. 4)[7,17]. We clearly show the early divergence of IK002 from the common ancestor of the other ancient and present-day East Eurasian and Native Americans (Fig. 1c). Given that the split between the East Asian lineage and the Northeast Asians/East Siberian and Native American lineage was estimated to be 26 kya[49], the divergence of the lineage leading to IK002 is likely to have occurred before this time but after 40 kya when the Tiányuán appeared (Fig. 4). Therefore, our results support the archaeological evidence based on lithic industry that the Jomon are direct descendants of the Upper-Paleolithic people who started living in the Japanese archipelago 38 kya (Supplementary Discussion).

We use the MA-1 ancestry as a proxy for ancestral populations who took the northern route of Himalaya mountains to come to East Eurasia. The fine stone tool, i.e., microblade, is a representative technology that was originally developed around Lake Baikal in Central Siberia during the Upper-Paleolithic period[9]. This microblade culture was also reached the Hokkaido island ~25 kya and the main island of the Japanese archipelago ~20 kya.

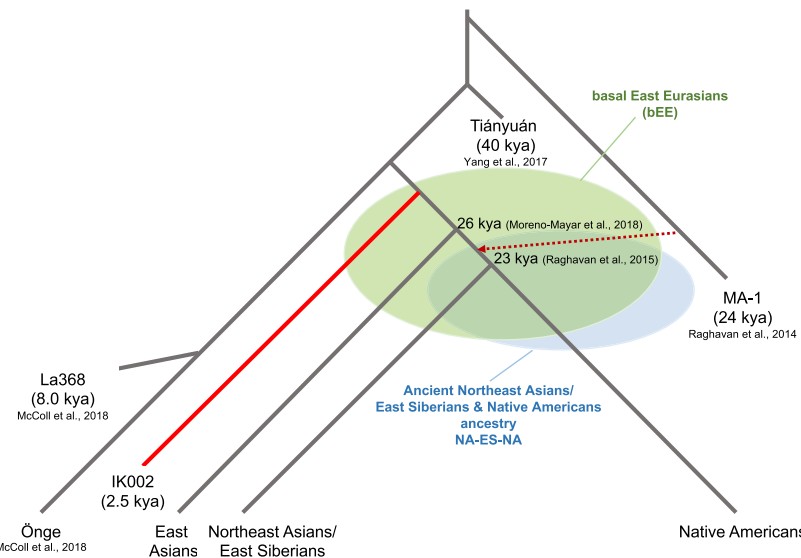

**Fig. 4 Schematic of peopling history in Southeast and East Asians, Northeast Asian/East Siberians and Native Americans.** The basal East Eurasians (bEE) are an ancient population that had no divergence among the ancestors of East Asians, Northeast Asians/East Siberian, and Native Americans. NA-ES-NA presents another ancient population that had no split between the ancestors of Northeast Asians/East Siberian and Native Americans.

If this culture was brought by demic diffusion, IK002 should still retain the MA-1 ancestry. However, we find no evidence on the genetic affinity of MA-1 with ancient and present-day Southeast/East Asians including Devil's Gate Cave (8.0 kya), Chokhopani (3.0–2.4 kya), and IK002 (2.5 kya) (Fig. 2). Therefore, we conclude that MA-1 gene flow occurred after the divergence between the ancestral populations of Northeast Asians/East Siberians/Native Americans (NA-ES-NA) and East Asians (Fig. 4): namely, East Asians originated in Southeast Asia without any detectable genetic influence from the ancestor who took the northern route. There are two hypothetical possibilities to explain the contradiction between genome data and archaeological records. The first possibility is that MA-1 may not be a direct ancestor who invented the microblade culture. The second is that, if the assumption is correct, then the northern-route culture represented by microblade was brought to the Japanese archipelago by the NA-ES-NA population who must have had substantial gene flow from MA-1, which was a typical "cultural diffusion." The first and second possibilities can be examined by obtaining genome data of Upper-Paleolithic specimens hopefully accompanying with microblade newly excavated from around Lake Baikal and from the Primorye region, respectively.

The genome of the Ikawazu Jomon (IK002) strongly supports the dual structure model of the origins of modern Japanese[37]. The PCA plot and phylogenetic tree showed that the present-day Japanese fell in the cluster of present-day East Asians (e.g., Han Chinese) but not clustered with IK002 (Fig. 1a & c), while a signal of gene flow was detected from IK002 to present-day Japanese (Fig. 1c & Supplementary Fig. 8). The PCA and ADMIXTURE showed the close relationship between IK002 and the Hokkaido Ainu even in the genome-wide structure reflected by linkage blocks (Supplementary Fig. 4 & Supplementary Fig. 6). These results fit the hypothesis that the Ainu and the Jomon share the common ancestor: the present-day Honshu Japanese are the hybrid between the Jomon and migrants from the East Eurasian continent, and the Hokkaido Ainu have less influence of genetic contribution of the migrants[37,42].

IK002 gave new insights into the migration route from south to north in East Eurasia. The $f_4$-statistics suggest that both the ancient and the present-day East Asians are closer to IK002 than Chokhopani (ancient Tibetans, 3.0–2.4 kya) in the coastal region but not in the inland region (Fig. 3 & Supplementary Fig. 9). Recent archaeological evidence shows little cultural influence from ancestral Austronesians to the Ryukyu islands (i.e., southernmost islands in the Japanese archipelago) in the last 10,000 years[51]. Given this cultural continuity, it is unlikely that this extra genetic affinity to IK002 was formed due to recent gene flow as shown in a previous study[52] (see Supplementary Discussion). Here, we provide two explanations for this signal: (1) the earliest-wave of migration from south to north occurred through the coastal region, and/or (2) the migration occurred in both the coastal and inland regions, but the genetic components of the earliest-wave were drowned out by back-migration(s) from north to south occurred in the inland region. In the early migration of anatomically modern humans, the route along the coast has been primarily thought to be important[3,53–56]. The use of water craft could support such explanation for the expansions through the islands and the coastal region[3], which supports the first explanation. There could be, however, potential criticisms: such archaeological evidence of craft boat is hardly found. Ulchi and Nivkh show significantly negative values of $f_4$-statistics ($Z = -4.541$ and $-10.148$, respectively. $p < 0.000006$). This could be an influence of the Hokkaido Ainu who are likely to be direct descendants of the Jomon people. The ancestor of the Ainu people could have admixed with the Okhotsk people[57] who were genetically as well as morphologically close to Ulchi and Nivkh in the the Primorye region[58–62]. The second explanation is that the track of the earliest-wave was erased in the inland but left over in the coastal region. Taiwan aborigines (Ami and Atayal) and Igorot are the Austronesian minorities. Taiwan aborigines are thought to have come from the East Eurasian continent $13.2 \pm 3.8$ kya[63], though the origin of Igorot (which can be derogatory, and could be best referred as Kankanaey) is not well-known.

The coastal-migration hypothesis would be supported by morphological traits of the Jomon individuals excavated from the Hobi shell-mound site, which is geographically very close to the Ikawazu shell-mound site in the Atsumi peninsula. Kaifu and Masuyama (2018) examined spatiotemporal variation in humeral shaft thickness using a sample of 1003 prehistoric individuals from various sites in the Japanese Archipelago, including 797 Jomon individuals[64]. The results showed that the Jomon humeri were thicker in coastal populations than in inland populations.

Kaifu and Masuyama (2018) hypothesized that this was caused by not only fishing in the outer sea, but also active marine transportation by rowing boats. This previous study suggests the Hobi Jomon individuals in the Atsumi peninsula were the people who adapted to coastal environment.

However, there is still lack of ancient genome data in East Asia critical to understand the peopling history of East Eurasians. Although our data support the idea that IK002 was the direct descendant of the Upper-Paleolithic people, how/where those Upper-Paleolithic people migrated to the Japanese archipelago remains unanswered. It is required to analyze more ancient genome data in order to fill the gap and to prove the hypothetical migration routes.

## Methods

**Human remains from archaeological sites in Jomon periods**. Twelve individuals from three archaeological sites (Ikawazu and Hobi Shell- mounds and Hegi Cave) were applied to this genomic study. The Ikawazu Shell-mound site locates in the Tahara city [34° 38′ 43″ north latitude; 137° 8′ 52″east longitude], Aichi Prefecture, where is in the central part of main island of the Japanese archipelago. The Hobi Shell-mounds, which are close to each other within ~5 km, locate on the center of main island (Honshu) (Supplementary Fig. 1). The Hegi Cave site locates on the northern part of Kyushu island. Based on conventional chronology of potteries, the individuals from the Hegi Cave site and the Hobi and Ikawazu Shell-mounds were assigned to the early to middle Jomon period (ca. 5000–8000 years ago) and the late to final Jomon periods (ca. 3500–2500 years ago), respectively (Supplementary Data 1).

**Archaeological information of the Ikawazu remains**. The Ikawazu Shell-mound was initially excavated in 1918[65]: over 100 individuals were excavated from the site, accompanied with the Jomon potteries assigned to the late–final Jomon period (ca. 3500–2500 years ago) based on the pottery chronology. More recently, one of us (T. M.) and colleagues excavated the new section within the Ikawazu Shell-mound site in 2010 and found six buried individuals of complete skeletal remains. IK001 and IK002 were recovered from Pit No.4, and showed a better state of preservation than those of the other remains. IK001 and IK002 had morphologically typical Jomon characteristics. On the side of the IK002 head, a Jomon pottery, so called a Gokan-no-mori type Jomon ware-corded which is typical in the late Jomon period, was offered. IK001 was excavated together with IK002: the former was an infant, and the latter was a late-middle-age woman. Preliminary PCR-direct sequencing of mitochondrial DNA (mtDNA) showed IK001 and IK002 had different mt D-loop sequences, suggesting they were not a mother–child relationship.

We extracted the collagen from IK001 and IK002, and obtained the purified gelatins for radiocarbon dating that were carried out using a compact AMS at The University Museum in The University of Tokyo. The conventional radiocarbon ages were estimated to 2638 ± 16 BP and 2681 ± 16 BP, respectively. Given that the Ikawazu people built the large shell-mound, it is likely that IK001 and IK002 had applied to marine resources. To correct the marine reservoir effect depending on intake ratio of marine fish and shell, we measured stable isotope ratios of carbon and nitrogen from extracted gelatins of IK001 and IK002. Calculating contribution of amino acids from marine resources with the two end-points of terrestrial herbivore and marine fish, the 50% marine were estimated. These data were calibrated by OxCal 4.3 (calibration program based on the calibration curve of IntCal 13)[66,67], and the calibrated ages of IK001 and IK002 showed 2699–2367 cal BP (95% CI) and 2720–2418 cal BP (95.4% CI), respectively. Because these ages were assigned to the Gokan-no-mori period which has no evidence of rice cultivation, we confirmed that IK001 and IK002 were individuals from the Jomon period accompanied with typical Jomon culture. We sampled teeth of IK001 (M1) and IK002 (M3), and fragments of the petrous bone of IK002.

**DNA extraction**. All the subsequent experiments of ancient DNA were performed in the clean room exclusively built for ancient DNA analyses installed in Department of Anatomy, Kitasato University School of Medicine. The bone and tooth pieces were cut by a sterile disc drill to separate crowns (enamel), roots (cementum and dentine) of teeth for all the samples, and the inner part of petrous bone only for IK002. DNA extraction was carried out with the protocol that is based on the previous studies[24,68] and modified in this study.

The tooth samples were cut by a sterile and UV-irradiated disc drill to separate crown (enamel) and root (cementum and dentine). DNA extraction of the root was carried out by the Gamba method with our modification. The teeth were washed by 3% sodium hypochlorite solution (Sigma-Aldrich) for 15 min, in order to decrease the degree of modern DNA contamination. After washing the teeth with ultrapure water (Thermo Fisher Scientific) and 99% ethanol (Sigma-Aldrich), the teeth were dried on the clean bench in the clean room for 16 h. The washed samples were pulverized by freezer mill (ShakeMaster Auto ver 2.0, BioMedical Science Inc.), and fine powder was obtained. To release DNA molecules from the sample powder,

200 mg tooth powder was incubated for 24 h at 55 °C followed by 24 h at 37 °C in 2 ml DNA LoBind tube (Eppendorf) with 1 ml lysis buffer in final concentrations of 20 mM Tris HCl (pH 7.5), 0.7% N-lauroylsarcosine, 47.5 mM EDTA (pH 8), 0.65 U/ml Proteinase K, shaking at 900 rpm in a Thermomixer (Eppendorf). The samples were then centrifuged at 13,000 g for 10 min and the supernatants were discarded. Fresh lysis buffer (1 ml) was added to the pellet, vortexed, and the incubation and centrifugation steps were repeated. The second supernatants were then transferred to ultrafiltration tubes (Amicon® Ultra-4 Centrifugal Filter Unit 30K, Merck), diluted with 3 ml TE (pH 8.0) and centrifuged at 2,000 g until final concentrations of ~100 ml were obtained. These volumes were then transferred to silica column (MiniElute PCR Purification Kit, QIAGEN) and purified according to manufacture's instructions, except for adding TWEEN 20 (at 0.05% final concentration) to 60 ul EB buffer pre-heated to 60 °C at the final step.

The petrous bone was cut by a sterile and UV-irradiated disc drill, and three pieces where Pinhasi et al. (2015) named as "C-part"[69] were obtained (C1, C2, C3); the pieces were washed by ultrapure water (Thermo Fisher Scientific) and 99% ethanol (Sigma-Aldrich). After the dried pieces were drilled and homogenized, ~500 mg bone powder was obtained from the three pieces. The first powder of 150 mg was used to extract DNA molecules following the modified protocol mentioned above. The powders of C1, C2, C3 were rinsed by ultrapure water [Rinsed supernatant], then treated with pre-digestion buffer containing 20 mM Tris HCl (pH 7.5), 0.7% N-lauroylsarcosine, 0.4 M EDTA (pH 8), 0.65 U/ml recombinant Proteinase K for 30 min at shaking at 900 rpm in a Thermomixer (Eppendorf). The mixture was then centrifuged at 13,000 g for 10 min and the supernatant was transferred to a 2 ml tube DNA LoBind [Pre-digestion]. Fresh lysis buffer (1 ml) containing 20 mM Tris HCl (pH 7.5), 0.7% N-lauroylsarcosine, 47.5 mM EDTA (pH 8), 0.65 U/ml recombinant Proteinase K was added to the pellet. After vortexed and incubated for 24 h at 55 °C followed by shaking at 900 rpm for 24 h at 37 °C, the first extract was obtained [Extract 1]. This step was then repeated, and the second extract [Extract 2] was obtained. The residual pellet was pulverized by wet-grinding with shaking sterile beads in grinding cylinder. Fresh lysis buffer containing 20 mM Tris HCl (pH 7.5), 0.7% N-lauroylsarcosine, 0.4 M EDTA (pH 8), 0.65 U/ml recombinant Proteinase K was added into the pulverized pellet, and the pellet was incubated for 24 h at 55 °C followed by shaking at 900 rpm for 24 h at 37 °C in 2 ml tube, the third extract was obtained [Extract 3]. The five elutes (rinsed and pre-digestion supernatants and three extracts) were filtrated following the protocol mentioned in the paragraph of DNA extraction from tooth. Finally, we obtained five DNA extracts from each petrous bone piece (total 15 extracts).

**Library construction**. DNA extracts obtained from these samples were quantified and qualified by Qubit 3.0 (Thermo Fisher Scientific) and Bioanalyzer (Agilent). Twenty-two extracts were used to construct 34 NGS libraries for Illumina sequencing system in Kitasato University. The NGS sequencing was carried out using MiSeq (Illumina) in Kyushu University and HiSeq (Illumina) in National Institute of Genetics in Japan. For cross-check, we separately prepared another five extracts from IK002, and made the NGS libraries in Copenhagen University and sequenced them using HiSeq in the Danish National High-Throughput DNA Sequencing Centre in Copenhagen.

Concerning C1 and C3, we used only one elute, [Extract 2], to construct NGS libraries and used to run on NGSs only in Japan. Meanwhile, concerning C2, we used five elutes, [Rinsed supernatant], [Pre-digestion], [Extract 1], [Extract 2], [Extract 3], and for constructing NGS libraries of C2, two different protocols were used separately in two laboratories (Kitasato University in Japan, and Copenhagen University Geogenetics Laboratory in Denmark) for inter-laboratory crosschecking. In the Kitasato University, the bead-based size selection protocol with the NEBNext Ultra DNA library preparation kit (New England Biolabs: NEB) was used. To remove large DNA fragments that could be contaminants from modern organisms, we modified the NEB original protocol: we adjusted the mixing ratio of the Agencourt AMPure XP solution (Beckman Coulter), the Solid Phase Reversible Immobilization magnetic bead solution. In the Copenhagen University, the protocol shown in Allentoft et al. (2015)[26] was used to make NGS libraries. Eventually, we constructed 6 libraries from tooth and 18 libraries from the petrous bone in the Kitasato University, and 5 libraries from the petrous bone in the Copenhagen University; totally we provided 29 libraries from IK002.

**Data output, processing, and authentication**. The 29 libraries were sequenced on a flowcell using the Ilumina HiSeq 2500 and the HiSeq reagent kit of normal and rapid mode for 100 cycles in paired end in the National Institute of Genetics in Japan and the Danish National High-Throughput DNA Sequencing Centre in Denmark. After running HiSeq, the sequence reads were called by Illumina's Real Time Analysis (RTA) or CASAVA ver. 1.8.2 (Illumina) base-calling software. The HiSeq output-data were processed using customizable NGS pipeline in the Geogenetics Laboratory and the Kitasato University. AdapterRemoval ver. 2[70] was used to trim adapters terminal N's (–trimns), low quality bases (-trim qualities,–minquality 2) and short reads (–minlength 30), and filtered reads were checked with FastQC ver. 0.11.7[71]. The filtered reads were mapped against hg19, human reference genome, by BWA ver. 0.5.9. Mapped reads with mapping quality below Phred score 30 and duplicates were removed using SAMtools[72] and the MarkDuplicates tool of Picard Tools (http://broadinstitute.github.io/picard/). Read depth and coverage were

determined using pysam (https://github.com/pysam-developers/pysam) and Bed-tools (https://github.com/arq5x/bedtools2).

Misincorporation patterns were assessed using mapDamage2[73]. The degree of modern DNA contamination was estimated by *ContamMix*[21] focused on mitogenome sequences. The resulting sequence assembly and haplotype for mitochondrial genome was visualized using MitoSuite ver. 1.0.9[74].

Analysis panels for population genetic inference were obtained by merging the mapped reads of IK002 and relevant previously published ancient individuals with two reference datasets of modern individuals:

- Panel "2240K": Genotypes for 404 whole-genome sequenced modern individuals[31,75–80], at 2,043,687 autosomal SNPs targeted for in-solution capture in previously published ancient DNA panels[81–83].
- Panel "HO Ainu": Genotypes for 2119 modern individuals from the Human Origins panel[84], as well as 36 Ainu individuals[85], at 41,264 SNPs overlapping between the two panels.

For both panels, pseudo-haploid genotypes for ancient individuals were generated by randomly sampling an allele passing filters (mapping quality ≥ 30 and base quality ≥ 30) at the reference panel SNP positions.

**Principal component analysis and *ADMIXTURE*.** As a first assessment of the genetic affinities of the study individuals we carried out PCA, as previously described[29,33,86]. In particular, we projected the low coverage ancient individuals onto the PCs inferred from different sets of modern and high coverage ancient individuals, using the 'lsqproject' and 'autoshrink' options in smartpca from the EIGENSOFT package[33], on both analysis panels. To explore shared genetic component between IK002, Ainu and the other populations, we ran ADMIXTURE ver. 1.3.0[44] on the "HO Ainu" panel. ADMIXTURE was run in ten replicates, for $K$ values ranging from $K = 2$ and $K = 20$, and best runs were selected and aligned using pong[87].

**TreeMix.** Maximum-likelihood trees and admixture graphs were inferred using *TreeMix*[46]. A subset of populations from the "2240K" panel were chosen to represent different ancestries of East Eurasians and Native Americans; IK002, East Asians (Han, Ami, Japanese and Devils Cave), Northeast Siberians (Lokomotiv and Shamanka, the ancient Siberians), Native Americans (Clovis and USR1, the ancestry of Native American), Himalayan (Sherpa, Kusunda and Chokhopani, the ancient highlander) and Southeast Asians (Önge and La368, the Hoabinhian). Furthermore, Tiányuán, Mal'ta (MA-1) and Ust'Ism were included as a landmark of divergence events happened in the Upper Paleolithic period. We used Mbuti as an outgroup and ran TreeMix with $m = 0$ to eight migration events to fit admixture graphs to the data. We only considered the SNP sites that are non-missing in all individuals included in this analysis and chose the tree under each condition that showed the highest likelihood among ten replicates with different random seeds.

***f*-statistics and *D*-statistics.** We used the *D*-statistic framework and $f_4$-statistical analyses to investigate patterns of admixture and shared ancestry in our data set. All *D*-statistics were calculated from allele frequencies using the estimators described previously[26,29], with standard errors obtained from a block jackknife (the jackknife parameter = 0.050, the number of blocks = 714). Calculating of *D*-statistics was carried out by qpDstat in the AdmixTools ver. 4.1[84]. The values of *D*-statistics were visualized and mapped by *R* software. $f_4$-statistics was calculated by qpDstat with the $f_4$ mode.

***ALDER* admixture dating.** To infer the timing of admixture we used ALDER[45] on the "HO Ainu" panel, for Japanese, Ainu and Ulchi as target populations. We used IK002 and the two Hokkaido Jomon individuals as a combined Jomon source, and Han, Ami, Korean or the ancient individuals from Devil's Gate cave as mainland East Asian source populations.

***qpGraph* modeling.** Admixture graph modeling with *qpGraph*[84] was carried out on the "2240K" panel. First, a backbone graph including ancient genomes representative of major divergences among East Asian lineages was fit: IK002 (early dispersal); Chokhophani (later dispersal, East Asia), and Shamanka (later dispersal, Siberia). Test populations of interest were then modeled as three-way mixtures of early (IK002) and later (Chokhopani, Shamanka) dispersal lineages, using a grid search in 1% increments of the two independent admixture proportions (using the 'lock' function in qpGraph).

**Statistics and reproducibility.** Ancient genomic data were generated using multiple libraries, which ensure reproducibility. Contamination ratios were <1%. All statistics was done using available packages and reproducibility can be accomplished using our own parameters mentioned in Methods.

**Reporting summary.** Further information on research design is available in the Nature Research Reporting Summary linked to this article.

## Data availability

This study has been evaluated by the Education Affairs in Tahara city in Aichi Prefecture, Japan. All genomic data (fastq formats) are available for download in the DNA DataBank of Japan (DDBJ) Sequence Read Archive (DRA. https://www.ddbj.nig.ac.jp/dra/index-e.html) under the accession numbers DRA005042 (Sample accession SAMD00058001) and at the European Nucleotide Archive (ENA) with accession number PRJEB26721 (Sample accession SAMEA4665869). The final bam datasets of mitochondrial- and nuclear-genome sequences, population genetic datasets of AdmixTools format (".indv", ".snp" and ".geno" files) and other source data are available on the server in University of Tokyo (https://drive.google.com/drive/folders/1IYZaq1WUbcP_ER2wWx224vP7rkX86NHw?usp=sharing). There is no any restrictions on data access in this study.

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

## Acknowledgements

The excavation of the Ikawazu Jomon individual was supported by Grant-in-Aid for Scientific Research (B) (25284157) to Y.Y. The Ikawazu Jomon genome project was organized by H.I., and T.H. & H.O. who were supported by MEXT KAKENHI Grant Numbers 16H06408 and 17H05132, by Grant-in-Aid for Scientific Research on Innovative Areas (Cultural History of Paleoasia), and by Grant-in-Aid for Challenging Exploratory Research (23657167) and for Scientific Research (B) (17H03738). The Ikawazu Jomon genome sequencing was supported by JSPS KAKENHI Grant Number 16H06279 to A.To., and partly supported in the CHOZEN project in Kanazawa University, and in the Cooperative Research Project Program of the Medical Institute of Bioregulation, Kyushu University. Computations for the Ikawazu Jomon genome were partially performed on the NIG supercomputer at ROIS National Institute of Genetics.

## Author contributions

H.O. initiated and led the study. T.G., S.N., R.K., Y.Y., H.I., E.W., M.S., and H.O. designed the study. H.I., T.H., and N.S. supervised morphological aspects of the project, and Y.Y. and H.St. supervised archaeological aspects of the project. H.O. and M.S. supervised the overall project. Y.Y., Y.M., H.St., S.M., O.K., and N.S. excavated, curated and described the skeletal remains from the Ikawazu and the Hobi shell-mound sites, and T.T., T.W., and H.I. did those from the Hegi cave site. T.G., R.S., and H.O. performed the sampling for DNA analysis, and M.Y. conducted radiocarbon dating. T.G., T.S., K.K., S.R., R.S., and M.A. performed the DNA laboratory work. H.Sb., A.Ta., and A.To., performed deep sequencing with high-throughput sequencers. T.G., S.N., M.S., S.R., T.K., B.N., H.M., and T.S. analyzed or assisted in the analysis of data. T.G., S.N., M.S., and H.O. interpreted the results with considerable input from M.Y., Y.Y. and H.I., and wrote the paper. All authors discussed the results and contributed to the final paper.

## Competing interests

The authors declare no competing interests.
