## [Peer Review File · Communications Biology]

Reviewers' comments:

Reviewer #1 (Remarks to the Author):

The manuscript focuses on genomic analysis of the 2.5 kya Jomon individual, IK002, an important piece of evidence that can inform us on the demographic history of eastern Eurasia. The authors concluded that IK002 is basal to all modern and ancient eastern Eurasians, and that, based the genetic affinity to Taiwanese aborigines, Jomon-related ancestry migrated into Japan from Southeast Asia likely via the coastal route. Overall the manuscript intends to provide a deeper investigation on the genetic origins and affiliations of IK002 relative to its earlier description in McColl et al., 2018 publication. However, the authors was unable to discuss and reconcile their current findings relative to the analysis already done in McColl et al., 2018 publication. In addition, the study was unable to provide standard population genetic tests (such as outgroup f_3 statistics, pairwise F_{st}) or provide statistical details on some of their analysis (qpGraph modelling) which are critically important in supporting their conclusions.

Specific comments:

1) Line 6 of the Abstract: 'IK002 forming a lineage to the rest of the ancient/present-day East Eurasians examined.' Eastern Eurasia can be defined as geographic region that encompasses East Asia, Southeast Asia, and Eastern Siberia, which will include ancient Hoabinhian individuals, Malay Negritos such as Jehai, Yana upper paleolithic individual, and Tianyuan. All of the aforementioned are more basal to the 'rest of East Eurasians.'

2) Paragraph 2, Line 12 of Introduction: 'This direct evidence on the link between the Jomon and Southeast Asians, thus, confirms the southern route origin of East Asians.' The evidence only shows genetic affinity between Jomon and Hoabinhian-related ancestry. Most ancient and present-day Southeast Asians and East Asians do not show strong genetic affiliations with Hoabinhian individuals. The finding can only support southern route origin of Jomon individual, and not all of East Asians. The populations of Southeast Asia and East Asia are already structured since the Paleolithic (Hoabinhian-related, Siberian-related, and ancestors of Austronesian-related and Austroasiatic-related populations), and some of this may have migrated from north of the Himalayan arc.

3) Paragraph 2, Line 7 of Results: 'We found that IK002 clusters between present-day Southeast and East Asians and the Upper-Paleolithic human remains (40 kya) from Tiányuán Cave.' Is the position of IK002 in the plot driven by Southeast Asian + East Asian versus Hoabinhian clusters, instead of Tianyuan? This is important point to look into, especially when IK002 is regarded as genetically affiliated with Hoabinhian-related ancestry.

4) Paragraph 2, Line 14 of Results: Hokkaido Ainu clearly clusters with IK002 based on PC analysis. Outgroup f_3 statistics must be performed to show whether Hokkaido Ainu shares the most drift with IK002. Also, pairwise F_{st} calculations between IK002 and all present day populations and ancient individuals must be included in the analysis.

5) Paragraph 3, Line 1. For Admixture analysis, how many iterations per K was ran? If multiple iterations was run, was CLUMPP applied after the Admixture? How consistent is the mode produced per K? Alternatively, a calculation of cross validation error per K can be performed. What is the basis for choosing $K = 10$ as the best representation for the population structure? It will be useful to plot all K 's in the supplementary.

6) Paragraph 4, Line 1 or Results. For Japanese and Ulchi as targets, what was the basis for Han Chinese as the surrogate source population? Was Ami, Atayal, Kankanaey, Dai, Korean and other east Asian populations tried? For Ainu as the target, was other combinations of source populations tried? An updated version of the software which can account for multiple pulses of admixture events (Malder) should be considered.

7) Paragraph 5, Line1 or Results. The basal position of IK002 relative to the selected East Asian and Native American groups can be attributed to genetic affiliation of IK002 to Hoabinhian-related ancestry, which was not accounted for in the Treemix run. Figure S38 of McColl et al., 2019 models IK002 as an admixture between Hoabinhian-related and East Asian related ancestry (represented by Ami). How do you reconcile the difference between the result of this treemix run and the earlier published qpGraph analysis? Would the topology be consistent if you include other present-day Southeast Asians, Siberian (Koryak, Yakut, etc), and Papuans in the Treemix analysis?

8) Paragraph 5, Lines 9-15 of Results. 'IK002 can be modelled as a basal lineage to East Asians, Northeast Asia/East Siberians, and Native Americans, supporting a scenario in which their ancestors arrived through the southern route and migrated from Southeast Asia towards Northeast Asia.' Given the comments above in items number 2 and 7, the conclusions in this section is not supported by robust analysis.

9) Paragraph 7, Line 7 of Results. Other than Ami, other populations such Kinh, Dusun, Kankanaey, etc, can be included in the analysis of combination D tests to support the conclusion. Instead of Ami, Papuans can also be used as a reference population, where one can examine whether any population X have gene flow with MA-1 relative to Papuan (which is known to have no MA-1-related ancestry).

10) Paragraph 8, Line 5 of Results. The signal of shared alleles between Ami/Atayal and IK002 might be due to subsequent migration of Austronesian-related populations into Japan, which can be consistent in the model presented in McColl et al., 2018 paper.

11) Paragraph 1, Line 2 of Discussion. As earlier presented, given the comments above in items number 2 and 7, the conclusion in this sentence is not supported by robust analysis.

12) Paragraph 2, Line 7. See comment 9 above.

13) Paragraph 4, Last sentence of Discussion. The word can be derogatory, and does not refer to a single ethnolinguistic population. They are best referred to as the 'Kankanaey' ethnic group.

14) Paragraph 5, Line 1 of Discussion. The important data on the worst f score of multiple f4 tests implement in qpGraph is not presented in the text nor in the figure S9. The worst f score can guide whether the models presented are rejected or not rejected. There is also no discussion as well on the discordance between the models presented in figure S9 of this paper and figure S28 of McColl et al 2018 paper.

Reviewer #2 (Remarks to the Author):

Key results: In this manuscript the authors report the whole-genome sequencing of an ancient individual from central Honshu, the main island of Japan in order to provide insights into the origin and migration history of East Asians. The specimen dates back to ~2500 BP and it is consider a

representative individual of the Jomon culture. Previous studies have been sequenced in depth other individuals (even older) of the Jomon culture, but from the northernmost island of Japan (Hokaido) and not from the main island of Japan. The main questions of the study are whether the Jomon were the direct descendants of the Upper Paleolithic first migrants into the Japanese archipelago and to determine if the Jomon culture is associated with the dispersal route from either the northern or the southern parts of Himalaya Mts. After sequencing the specimen genome at a considerable depth (~1.85X) for aDNA standards regarding the expected preservation at humid and warm climates, they performed several population genetics methods and found that, indeed, this individual forms a lineage basal to the rest of the ancient or present-day East Eurasians examined in this study and that the route of migration that this lineage followed was toward East Asia from Southeast Asia.

Validity: No major flaws were found to prohibit the publication of the manuscript. However, revisions are needed, as shown in the next sections in more detail.

Originality and significance: The originality and significance are obvious as in most ancient DNA studies that examine the genome of missing links and representative individuals of culture around the world. However, this individual has been sequenced in the same depth at 1.85X in a previous study (McColl et al. 2018 Science DOI: 10.1126/science.aat3628) and I guess the wetlab part is the same per se for the both of them. Thus, the wetlab part (DNA extraction, libraries etc) is not original, despite that the authors provide many details in contrast to the 2018 paper that just cites a published protocol. This means that in this paper the authors use previously published genome data despite being the same team who published the 2018 paper. On the other hand, the 2018 paper does not mention anything about data availability. Thus, I have to rise my skepticism if this paper should be considered original in the term of generating new genome data. Maybe it should be made clear to the reader that the genomic data presented here are coming from a previous study and that in this paper the authors provide detailed M&M regarding this specimen. The bioinformatics analyses, though, are indeed original, as the authors use a different dataset and also address different scientific questions.

Data & methodology: In this part I will focus on the reproducibility of the analyses taking into account the supplementary info, too. The wetlab part is properly described. However, the sequence data are not publicly available (nothing is mentioned in this manuscript about data availability) so no one can replicate the analyses. I suggest to inform if they going to be uploaded in a public database or if can someone request them directly from the authors. Also, the data processing and the population genomics analyses are not well-documented in order to an independent scientist could replicate the results. This part is very quickly written, in some parts is very confusing, and in some cases the authors just mention the software tool without any detail on what parameters they used (or if they just used the default) neither are citing any particular published pipeline that they followed in order to not have to provide such details. This entire part needs revision (I provide some more details below). Moreover, in a few cases the version of the tools is omitted including but not limited to samtools, plink and treemix (different version may result in different output).

Appropriate use of statistics and treatment of uncertainties: In my opinion the statistical analyses, wherever provided with details, are robust. The bar with migration plot from treemix graph in Fig.1 is missing, though.

Conclusions: The conclusions are valid regarding the interpretation of the population genomics results and their association with the archaeological background of the samples and the hypotheses made.

References: There are some missing citations mostly for software tools used; they are given only at the supplementary info and even there, some of them they do not have any citation e.g. bedtools.

Clarity and context: The main text is clear and appropriate, albeit there is erroneous link between the supplementary info (e.g. Text S2 should be Text S1 etc). The language needs to be edited a bit by a native English speaker, especially the Supplementary Info text. Attention should be given as there are a few spaces missing (maybe due to PDF conversion?). The length of the main manuscript is acceptable.

Out of my expertise: I cannot evaluate the technical details for the carbon dating analyses.

Suggested improvements: Other than mentioned above, here I provide some more specified details.

Note> It would have been helpful for the reviewers to provide a line numbers and speed up in this way the process of finding the text part that is mentioned here.

Main Text

Results

Ancient DNA prescreening, dating, and sequencing by high-throughput sequencers

1. Line 6: It is not clear why Text S2 is given here. I cannot find a connection.
2. The method/algorithm the authors used for degerming the genetic sex is not given in the M&M. Testing whether IK002 is the direct descendant of the Upper Paleolithic people in the Japanese archipelago

1. There is a part that you explain how the authors merged previously published data with the IK002 data. I propose some details to be moved to the M&M section.
2. The ALDER analysis is not mentioned at all in the M&M section.

Materials

1. In Line 7 the link to Table S1 is not provide any information associated to the text. Maybe the authors mean Text S1?

Methods

1. 34 NGS libraries: 24? (18 teeth and 6 petrous)
2. Please provide the company of Miseq etc as you did with other instruments e.g. Bioanalyzer
3. Since the HGDP is mentioned for the first time please provide a reference.
4. Please provide here or in the supplementary info a rationale of selecting 1 to 6 possible migration events.

Fig. 3 Legend

1. Do you mean "Triangle label means strong/high statistical significance"?

References

1. Please check the Kanzawa-Kiriyama et al. 2019 reference (advpub?)
2. Please ensure that the format is the same for every reference. For example there a few ones that the journal abbreviation does not have dots.

Supplementary Info

Table S2

1. Please inform the reader what do the terms endogenous and efficiency are expressing. How can someone calculate them?

Text S2

1. In some case the μ l are given as ml e.g. 60 ml EB (perhaps a PDF conversion fault).
2. In the part where there are different petrous parts used (C1, C2, C3) it is not clear which one did you used for NGS libraries, with what method and in which lab.

You mention We used [Extract2] of C1 and C3 to construct NGS libraries.

And then below: Two different protocols of construction NGS libraries from five elutes of C2 ([Rinsed supernatant], [Pre-digestion], [Extract 1], [Extract 2], [Extract 3])...

This suggest this part to be written in order to be less confusing.

3. The part from "To remove large DNA fragment..." to "...for DNA solution" is very confusing. Please rephrase.

Text S3

1. Where can someone find this customizable NGS pipeline that is mentioned? Is it described below or in another paper perhaps? Are any in-house scripts used and not given to the reader in a way that he will not be able to replicate the results? Please elaborate.
2. Some references are missing e.g. BWA and others.
3. Please ensure that in all the cases that you are mention the version of a software tool (here and in the main text) are following the same format (e.g. ver. 1.0.0 vs v1.0.0 vs -1.0.0).

Text S4

1. I think you should mention in every case after you merged IK002 with the different genome databases how many SNPs had in common. Are all the analyses using the same number of SNPs? If

not please inform the reader when the number changes.

2. Please provide a rationale to the reader to help understand why did you select the Ainu people in present-day populations and the Chokhopani 1 and Tiányuán individuals in ancient populations in ADMIXTURE analysis.

3. The first sentence in the TreeMix paragraph is confusing. Please rephrase.

Text S5

1. This entire text section is not linked to the main article. Maybe somewhere in the Discussion?

References

1. The format here is not the same with the one used in the main article.

2. Baba et al. 1991: Can the journal name be abbreviated?

Reviewer #3 (Remarks to the Author):

The manuscript "Jomon genome sheds light on East Asian population history" by Gakuhari et al. describes production and analysis of genome-wide data from a ~2.5 kya individual from Honshu, Japan, associated with the hunter-gatherer-fisher Jomon culture that is hypothesized to represent the descendants of the earliest Upper Paleolithic settlers of the Japanese archipelago.

This represents the first Jomon genome with appreciable coverage from mainland Japan, although a low coverage Jomon genome from Honshu and two higher coverage ones from Hokkaido have been published previously. Analysis of this individual, IK002, albeit at lower coverage, has been published earlier in McColl et al. 2018.

The main results of the study are that:

1) Jomon represent an early diverging lineage basal to the split of Tibetans and the ancestors of other East Asians, Eastern Siberians and Native Americans, but dating after the split of the lineage of which Tianyuan (40kya) is a part of. As split of the ancestors of East Asians and Native Americans is dated to around 26kya, the window of divergence is consistent with Jomon being the descendants of Upper Paleolithic settlers. This is consistent with the findings of Kanzawa-Kiriyama et al. 2019.

2) The ancestors of Jomon took a southern route from South East Asia to reach the archipelago (consistent with Kanzawa-Kiriyama et al. 2019) and the Jomon shows no evidence of admixture with representatives of a northern route (here represented by the Siberian MA1). Similarities in microblade technology across northern Eurasia and in Japan in the UP could have arisen through cultural diffusion or demic diffusion by a group not related to MA1.

3) Jomon show genetic affinity to modern and ancient coastal, but not inland, populations of East Asia, possibly supporting a coastal route of spread for this early ancestry.

4) As previously described, modern Ainu are largely direct descendants of the Jomon with limited recent Japanese (East Asian-derived) gene-flow (Kanzawa-Kiriyama et al. 2019), while Japanese have a smaller fraction of Jomon ancestry (McColl et al. 2018).

The wet lab processes and computational analyses are sound and according to best practices. The argumentation is easy to follow and supported by the results. The authors do not overstate their conclusions and present alternative hypotheses where appropriate, also taking into account archaeological and anthropological evidence.

The weakness of the paper is twofold: First, limiting the analysis to only the IK002 Jomon genome while other Jomon genomes have been published. This might be due to the very recent publication of these and lack of time to redo and add analyses. For a resubmission I suggest including the additional Jomon genomes which could also address further questions of Jomon population history, e.g. overall population size, population size differences in the different islands, contact between islands and between islands and mainland, dating/location of gene-flow into modern Japanese etc.

This would help with the second weakness, the large overlap with recent publications. Additional analyses could set the paper apart from previously published ones, especially McColl et al. 2018 and Kanzawa-Kiriyama et al. 2019, as many of the findings here are not exactly novel or surprising. Furthermore, the authors could address and explain differences in their findings to the previous papers, such as different admixture proportions of Jomon into Ainu and modern Japanese, or the finding in McColl that "the Jōmon individual is best-modeled as a mix between a population related to group 1/Önge and a population related to East Asians" rather than a deep diverging lineage.

- Concerning the Jomon-like ancestry in Ulchi and Nivkh, to distinguish the scenarios "This could be an influence of the Hokkaido Ainu who are likely to be direct descendants of the Jomon people. The ancestor of the Ainu people could have admixed with the Okhotsk people⁵⁵ who were genetically as well as morphologically close to Ulchi and Nivkh in the the Primorye region^{56–60}. The second explanation is that the track of the earliest-wave was erased in the inland but left over in the coastal region.", can you use ALDER to date the admixture event? Same for the Jomon-like ancestry in Devil's Gate. Are these events independent?

Suggested edits and small mistakes that should be fixed:

-mention the geographic origin of the sample in the abstract

-Some of the results are referenced for the first time in the discussion, such as the qpgraph modeling. These should get their own paragraph in the results.

-Fig. 1: The labeling in B is hardly legible; Loschbour is a Mesolithic individual from West Eurasia not Central Siberia

-Fig. 4 needs more explanation in the caption. What is bEE, a population that admixed with the ancestors of IK002, East Asians and NEAsians/East Siberians, or an encompassing term to describe the basal lineage from which the lineages leading to IK002, East Asians and NEAsians/East Siberians branched off?

-Fig. S4: Add a separate legend. Jomon samples are called F23 and F5 in caption, but fun23 and fun5 in plot

-Fig. S7: duplicated sentence in caption: "The scale bar shows the average standard error (SE) of the entries in the covariance matrix."

-Fig. S8: A and B switched in caption; Ust'Belaya mapped in wrong location, should be near Lake Baikal

-The texts especially of the supplementary material could use a round of copyediting, there are numerous typos or grammatical mistakes, e.g. in Materials and Methods: "we chose IK002, who were excavated from" ("were" > "was"). The main text looks mostly fine.

-As a visual summary of the results, a schematic such as Fig. 4 in McColl et al. 2018 (but focusing on the settlement waves of Japan and dispersal of Jomon-like ancestry) would be a nice addition, especially since this study deals with routes of dispersal.

Reviewers' comments:

Reviewer #1 (Remarks to the Author):

The manuscript focuses on genomic analysis of the 2.5 kya Jomon individual, IK002, an important piece of evidence that can inform us on the demographic history of eastern Eurasia. The authors concluded that IK002 is basal to all modern and ancient eastern Eurasians, and that, based the genetic affinity to Taiwanese aborigines, Jomon-related ancestry migrated into Japan from Southeast Asia likely via the coastal route. Overall the manuscript intends to provide a deeper investigation on the genetic origins and affiliations of IK002 relative to its earlier description in McColl et al., 2018 publication. However, the authors was unable to discuss and reconcile their current findings relative to the analysis already done in McColl et al., 2018 publication. In addition, the study was unable to provide standard population genetic tests (such as outgroup f3 statistics, pairwise Fst) or provide statistical details on some of their analysis (qpGraph modelling) which are critically important in supporting their conclusions.

Specific comments:

1) Line 6 of the Abstract: 'IK002 forming a lineage to the rest of the ancient/present-day East Eurasians examined.' Eastern Eurasia can be defined as geographic region that encompasses East Asia, Southeast Asia, and Eastern Siberia, which will include ancient Hoabinhian individuals, Malay Negritos such as Jehai, Yana upper paleolithic individual, and Tianyuan. All of the aforementioned are more basal to the 'rest of East Eurasians.'

Answer to comment 1)

Thank you for the reviewer's comment. We agree with the comment and the sentence can be misleading, and changed the sentence as below:

'IK002 forming a basal lineage to the ancient/present-day East and Northeast Asians examined.'

2) Paragraph 2, Line 12 of Introduction: 'This direct evidence on the link between the Jomon and Southeast Asians, thus, confirms the southern route origin of East Asians.' The evidence only shows genetic affinity between Jomon and Hoabinhian-related ancestry. Most ancient and present-day Southeast Asians and East Asians do not show strong genetic affiliations with Hoabinhian individuals. The finding can only support southern route origin of Jomon individual, and not all of East Asians. The populations of Southeast Asia and East Asia are already structured since the Paleolithic (Hoabinhian-related, Siberian-related, and ancestors of Austronesian-related and Austroasiatic-related populations), and some of this may have migrated from north of the Himalayan arc.

Answer to comment 2)

Thank you so much for the reviewer's comments. Yes, the sentence could be misleading. We changed the sentence as below:

'This direct evidence on the link between the Jomon and Southeast Asians, thus, confirms the southern route origin of the IK002 lineage.'

3) Paragraph 2, Line 7 of Results: 'We found that IK002 clusters between present-day Southeast and East Asians and the Upper-Paleolithic human remains (40 kya) from Tiányuán Cave.' Is the position of IK002 in the plot driven by Southeast Asian + East Asian versus Hoabinhian clusters, instead of Tianyuan? This is important point to look into, especially when IK002 is regarded as genetically affiliated with Hoabinhian-related ancestry.

Answer to comment 3)

Thank you so much for the reviewer's comments. In the PCA, IK002 clusters between Southeast Asian + East Asian modern populations and Tianyuan / Hoabinhian, albeit closer to Tianyuan. The position on the PCA alone is not sufficient information to distinguish the two scenarios the reviewer mentions. To avoid the sentence being misleading we have changed it as below.

'We found that IK002 clusters between present-day Southeast and East Asians and ancient Hoabinhian hunter-gatherers and the Upper-Paleolithic (40 kya) individual from Tiányuán Cave

4) Paragraph 2, Line 14 of Results: Hokkaido Ainu clearly clusters with IK002 based on PC analysis. Outgroup f_3 statistics must be performed to show whether Hokkaido Ainu shares the most drift with IK002. Also, pairwise F_{ST} calculations between IK002 and all present day populations and ancient individuals must be included in the analysis.

Answer to comment 4)

Thank you so much for the reviewer's comments. We have added outgroup f_3 statistics for IK002 and Ainu as supplementary figure S5 and table S4. The new results show that Ainu share most drift with the ancient Jomon individuals (Hokkaido and IK002), as well as that IK002 shares most drift with Hokkaido Jomon and Ainu. We have added a sentence referring to these results:

'Outgroup f_3 statistics support the results from the PCA clustering, with IK002 sharing most genetic drift with the Hokkaido Jomons, followed by the Ainu (Fig. S5, Table S4).'

Pairwise F_{ST} calculations using IK002 can not be carried out with pseudo-haploid genotypes as used to represent IK002.

5) Paragraph 3, Line 1. For Admixture analysis, how many iterations per K was ran? If multiple iterations was run, was CLUMPP applied after the Admixture? How consistent is the mode produced per K? Alternatively, a calculation of cross validation error per K can be performed. What is the basis for choosing K = 10 as the best representation for the population structure? It will be useful to plot all K's in the supplementary.

Answer to comment 5)

Thank you for the reviewer's comment. We have updated the description and plots of the admixture analysis accordingly. ADMIXTURE was run in 10 replicates, for K values ranging from K=2 and K=20, and best runs were selected and aligned using pong. For this revised version, we chose to represent the structure at K=15 to highlight fine-scale structure among different modern and ancient groups from Eastern Asia, including a component maximized in the Jomon individuals We also provide all runs from K=2 to K=20 in the supplement as figure S6 for the interested reader.

6) Paragraph 4, Line 1 or Results. For Japanese and Ulchi as targets, what was the basis for Han Chinese as the surrogate source population? Was Ami, Atayal, Kankanaey, Dai, Korean and other east Asian populations tried? For Ainu as the target, was other combinations of source populations tried? An updated version of the software which can account for multiple pulses of admixture events (Malder) should be considered.

Answer to comment 6)

Thank you for the reviewer's comment. Yes, we used Han Chinese as the surrogate source population, because the robust archaeological evidence showed migrations accompanying rice cultivation and horse-breeding culture came to the Ikawazu area from China (probably through the Korean peninsula) after 2.4 kya (see supplementary info. Text S1). We nevertheless updated the ALDER analysis in the revised manuscript, to include Ami, Korean and the ancient individuals from Devil's Gate cave as additional source populations, for

Japanese, Ainu and Ulchi as targets. We also now combined IK002 and the two Hokkaido Jomon individuals into a combined Jomon source population to increase power of the analysis. The results of these analyses show a slightly earlier estimated admixture time (60-76 generations ago), but consistent across different choices of source populations for mainland East Asian ancestry. Results are provided in the updated supplementary table S5. We did not use MALDER to test for additional admixture pulses from another source population for two reasons: 1) the archaeological evidence does not suggest additional major migrations after the introduction of rice cultivation, and 2) the exponential curve fits for the two reference population models show a good fit to the observed weighted LD patterns. We provide those plots now as additional supplementary figures S8, and have updated the section of the manuscript accordingly.

7) Paragraph 5, Line1 or Results. The basal position of IK002 relative to the selected East Asian and Native American groups can be attributed to genetic affiliation of IK002 to Hoabinhian-related ancestry, which was not accounted for in the Treemix run. Figure S38 of McColl et al., 2019 models IK002 as an admixture between Hoabinhian-related and East Asian related ancestry (represented by Ami). How do you reconcile the difference between the result of this treemix run and the earlier published qpGraph analysis? Would the topology be consistent if you include other present-day Southeast Asians, Siberian (Koryak, Yakut, etc), and Papuans in the Treemix analysis?

Answer to comment 7)

Thank you for the reviewer's comment. In their study, McColl et al also performed treemix analyses for IK002 (Fig. S29 in their paper) with a similar set of populations as well as Papuans and Denisova. The results from their analysis are consistent with those presented here, with IK002 as a basal lineage, and no apparent admixture signal from Hoabinhian hunter-gatherers. We note that with a complex admixture graph model as the one in McColl et al it is unfeasible to

explore the full space of topologies, and hence equally or better fitting models might not have been explored. In particular, McColl added IK002 to an admixture graph which already included Ami, thereby not allowing for Ami to be modeled as admixed between Jomon and another Asian lineage. The Jomon-like ancestry in the Ami we demonstrate here could therefore cause the apparent Ami/Hoabhinian admixture for IK002 in their model, in order to account for the excess shared drift between IK002 and Ami. In conclusion, we are confident in our results of IK002/Jomon representing a basal East Asian lineage, but caution that more ancient samples will be needed to disentangle the more complex relationships among those early lineages.

8) Paragraph 5, Lines 9-15 of Results. 'IK002 can be modelled as a basal lineage to East Asians, Northeast Asia/East Siberians, and Native Americans, supporting a scenario in which their ancestors arrived through the southern route and migrated from Southeast Asia towards Northeast Asia.' Given the comments above in items number 2 and 7, the conclusions in this section is not supported by robust analysis.

Answer to comment 8)

Thank you for the reviewer's comment. As commented in the reply to point (7) above, we disagree with the reviewer doubting the robustness of IK002 lineage as a basal position. In both McColl et al and here, IK002 consistently forms a basal lineage in the Treemix analysis, diverging prior to ancestors of modern East Asians, Northeast Asia/East Siberians, and Native Americans. We also present a formal test of the basal position using D-statistics, which are consistent with IK002 forming an outgroup to pairs of East and Northeast Asian populations and the ancient Tibetan individual from Chokhopani (Fig. 3 and Fig. S8). We therefore argue that our conclusions and this statement are warranted. To clarify the dual origin of Native Americans as previously described, we have modified the phrase to now state

“IK002 can be modelled as a basal lineage to East Asians, Northeast Asia/East Siberians, and Native Americans, supporting a scenario in which their ancestors arrived through the southern route and migrated from Southeast Asia towards Northeast Asia. However, we note that Siberian and Native American populations also harbor differing levels of West Eurasian-related ancestry (represented by the Upper Paleolithic individual from Mal’ta), which likely arrived through a northern route. “

9) Paragraph 7, Line 7 of Results. Other than Ami, other populations such Kinh, Dusun, Kankanaey, etc, can be included in the analysis of combination D tests to support the conclusion. Instead of Ami, Papuans can also be used as a reference population, where one can examine whether any population X have gene flow with MA-1 relative to Papuan (which is known to have no MA-1-related ancestry).

Answer to comment 9)

Thank you for the reviewer's comment. In the D-tests described in this section (shown in figure 2), we used Ami as an East Asian reference population to quantify excess allele sharing for all other Asian populations and ancient individuals in our dataset. As seen in the figure, all East and Southeast Asian populations, including Dusun and Kinh are consistent with forming a clade with Ami, i.e. do not show any evidence of excess allele sharing with MA1. As such, replacing Ami with Kinh or Dusun is not expected to have any effect on those statistics. We did not use Papuan as the Denisovan ancestry in Papuans would lead to excess allele sharing with the outgroup (Mbuti), and hence systematically skew the statistics.

10) Paragraph 8, Line 5 of Results. The signal of shared alleles between Ami/Atayal and IK002 might be due to subsequent migration of Austronesian-

related populations into Japan, which can be consistent in the model presented in McColl et al., 2018 paper.

Answer to comment 10)

Thank you for the reviewer's comment. We agree with the reviewer that Austronesian-related gene-flow into Japan could in principle also cause this signal. We do however find it to be a very unlikely scenario, as there is no evidence in the archaeological record for an Austronesian migration into Japan. Furthermore, Ami are consistent with being an outgroup to IK002 and the Hokkaido Jomon when using $f_4(\text{Mbuti}, \text{Ami}; \text{IK002}, \text{Hokkaido Jomon})$, with $Z = -1.4$ (Table S3). Hence any potential gene flow from Austronesians groups would have to have had a similar impact across a large area of the Japanese archipelago, as well as reaching northern Japan by at least 3,500 years ago (age of the Hokkaido Jomon samples). In conclusion we don't find support for this alternative interpretation

11) Paragraph 1, Line 2 of Discussion. As earlier presented, given the comments above in items number 2 and 7, the conclusion in this sentence is not supported by robust analysis.

Answer to comment 11)

Thank you for the reviewer's comment. As we answered to your comments 2, we changed the sentence to avoid mislead as below .

"IK002 is modelled as a basal lineage to East Asians, Northeast Asians/East Siberians, and Native Americans (basal East Eurasians: bEE) after the divergence between Tianyuan and the ancestor of hunters-gatherers in Southeast Asia."

12) Paragraph 2, Line 7. See comment 9 above.

Answer to comment 12)

Thank you for the reviewer's comment. We already answered to your comment 9.

13) Paragraph 4, Last sentence of Discussion. The word can be derogatory, and does not refer to a single ethnolinguistic population. They are best referred to as the 'Kankanaey' ethnic group.

Answer to comment 13)

Thank you for the reviewer's comment. However, the data are derived from Human Genome Diversity Project. If we change the name of the population, researchers cannot recognize them. So, we added some words into the sentence as below.

"though the origin of Igorot (which can be derogatory, and could be best referred as Kankanaey) is not well known."

14) Paragraph 5, Line 1 of Discussion. The important data on the worst f score of multiple f4 tests implement in qpGraph is not presented in the text nor in the figure S9. The worst f score can guide whether the models presented are rejected or not rejected. There is also no discussion as well on the discordance between the models presented in figure S9 of this paper and figure S28 of McColl et al 2018 paper.

Answer to comment 14)

We thank the reviewer for the comment. We note that the aim of the qpGraph analysis was to quantify the range of possible Jomon-like admixture in different target populations, including Ami, and hence the same considerations with regards to the results of McColl et al discussed above apply as well. To improve clarity of the results, we have now included the worst fitting Z-score for all models in figure S10, as well as the fit of base model without the target admixed

population. As can be seen in the base model, when performing the initial qpGraph fit only on ancient groups and excluding Ami, a model with Jomon as basal lineage and no contribution from an Onge/Hoabinian-like group provides a very good fit to the data (worst $Z=2.1$).

Reviewer #2 (Remarks to the Author):

Key results: In this manuscript the authors report the whole-genome sequencing of an ancient individual from central Honshu, the main island of Japan in order to provide insights into the origin and migration history of East Asians. The specimen dates back to ~2500 BP and it is considered a representative individual of the Jomon culture. Previous studies have been sequenced in depth other individuals (even older) of the Jomon culture, but from the northernmost island of Japan (Hokaido) and not from the main island of Japan. The main questions of the study are whether the Jomon were the direct descendants of the Upper Paleolithic first migrants into the Japanese archipelago and to determine if the Jomon culture is associated with the dispersal route from either the northern or the southern parts of Himalaya Mts. After sequencing the specimen genome at a considerable depth (~1.85X) for aDNA standards regarding the expected preservation at humid and warm climates, they performed several population genetics methods and found that, indeed, this individual forms a lineage basal to the rest of the ancient or present-day East Eurasians examined in this study and that the route of migration that this lineage followed was toward East Asia from Southeast Asia.

Validity: No major flaws were found to prohibit the publication of the manuscript. However, revisions are needed, as shown in the next sections in more detail.

Originality and significance: The originality and significance are obvious as in most ancient DNA studies that examine the genome of missing links and representative individuals of culture around the world. However, this individual has been sequenced in the same depth at 1.85X in a previous study (McColl et al. 2018 Science DOI: 10.1126/science.aat3628) and I guess the wetlab part is the

same per se for the both of them. Thus, the wetlab part (DNA extraction, libraries etc) is not original, despite that the authors provide many details in contrast to the 2018 paper that just cites a published protocol. This means that in this paper the authors use previously published genome data despite being the same team who published the 2018 paper. On the other hand, the 2018 paper does not mention anything about data availability. Thus, I have to rise my skepticism if this paper should be considered original in the term of generating new genome data. Maybe it should be made clear to the reader that the genomic data presented here are coming from a previous study and that in this paper the authors provide detailed M&M regarding this specimen. The bioinformatics analyses, though, are indeed original, as the authors use a different dataset and also address different scientific questions.

Data & methodology: In this part I will focused on the reproducibility of the analyses taking into account the supplementary info, too. The wetlab part is properly described. However, the sequence data are not publicly available (nothing is mentioned in this manuscript about data availability) so no-one can replicate the analyses. I suggest to inform if they going to be uploaded in a public database or if can someone request them directly from the authors. Also, the data processing and the population genomics analyses are not well-documented in order to an independent scientist could replicate the results. This part is very quickly written, in some parts is very confusing, and in some cases the authors just mention the software tool without any detail on what parameters they used (or if they just used the default) neither are citing any particular published pipeline that they followed in order to not have to provide such details. This entire part needs revision (I provide some more details below). Moreover, in a few cases the version of the tools is omitted including but not limited to samtools, plink and treemix (different version may result in different output).

Appropriate use of statistics and treatment of uncertainties: In my opinion the statistical analyses, wherever provided with details, are robust. The bar with migration plot from treemix graph in Fig.1 is missing, though.

Conclusions: The conclusions are valid regarding the interpretation of the population genomics results and their association with the archaeological background of the samples and the hypotheses made.

References: There are some missing citations mostly for software tools used; they are given only at the supplementary info and even there, some of them they do not have any citation e.g. bedools.

Clarity and context: The main text is clear and appropriate, albeit there is erroneous link between the supplementary info (e.g. Text S2 should be Text S1 etc). The language needs to be edited a bit by a native English speaker, especially the Supplementary Info text. Attention should be given as there are a few spaces missing (maybe due to PDF conversion?). The length of the main manuscript is acceptable.

Out of my expertise: I cannot evaluate the technical details for the carbon dating analyses.

Suggested improvements: Other than mentioned above, here I provide some more specified details.

Note> It would have been helpful for the reviewers to provide a line numbers and speed up in this way the process of finding the text part that is mentioned here.

Main Text

Results

Ancient DNA prescreening, dating, and sequencing by high-throughput sequencers

1. Line 6: It is not clear why Text S2 is given here. I cannot find a connection.

Thank you for the reviewer's comment. Because this is a simple mistake, we removed "Text S2" in Line 6. We noticed Text S2 in Line 3 should also be Text S1.

2. The method/algorithm the authors used for degerming the genetic sex is not given in the M&M.

Thank you for the reviewer's comment. We added the sentence as below.

"We determined the genetic sex following the method of Fu et al. 2013. The ratio of X and Y chromosome was 3:1, suggesting the specimen is female. This agree with the morphological determination of the sex."

Testing whether IK002 is the direct descendant of the Upper Paleolithic people in the Japanese archipelago

1. There is a part that you explain how the authors merged previously published data with the IK002 data. I propose some details to be moved to the M&M section.

Thank you for the reviewer's comment. We wanted to provide some more detail on the merged dataset for the results section here, but, if the reviewer and editor strongly recommend removing the explanation, we would be ok with moving it to the methods section

2. The ALDER analysis is not mentioned at all in the M&M section.

Thank you for the reviewer's comment. We added the ALDER in the M&M section.

Materials

1. In Line 7 the link to Table S1 is not provide any information associated to the text. Maybe the authors mean Text S1?

Thank you for the reviewer's comment. Yes, it is not Table S1 but Text S1.
We changed it.

Methods

1. 34 NGS libraries: 24? (18 teeth and 6 petrous)

Thank you for the reviewer's comment. The details are described in Paragraph 3 of Text S2.

2. Please provide the company of Miseq etc as you did with other instruments e.g. Bioanalyzer

Thank you for the reviewer's comment. Yes, we added the company name, Illumina, in the text.

3. Since the HGDP is mentioned for the first time please provide a reference.

Thank you for the reviewer's comment. Yes, we added the reference in the text.

4. Please provide here or in the supplementary info a rationale of selecting 1 to 6 possible migration events.

Thank you for the reviewer's comment. When using $m=8$ migration events, the worst residuals of the treemix fit are within 3.3 standard errors. We chose to not include more migration events to avoid overfitting and difficulties with interpretations of the resulting complex graphs. We updated Fig. S9 to now also include the matrix of residuals.

Fig. 3 Legend

1. Do you mean "Triangle label means strong/high statistical significance"?

Thank you for the reviewer's comment. We changed the sentence as below.

"The shape represents statistical significances of genetic affinities based on Z score. Triangle label means strong/high statistical significance with $|Z|>3$, inverted triangle means weak significance with $|Z|=2\sim3$ and circle means non-significance with $|Z|<2$."

References

1. Please check the Kanzawa-Kiriyama et al. 2019 reference (advpub?)

Thank you for the reviewer's suggestion. We changed it as below.

"Kanzawa-Kiriyama, H. et al. Late Jomon male and female genome sequences from the Funadomari site in Hokkaido, Japan. *Anthropol. Sci.* 127, 83-108 (2019)."

2. Please ensure that the format is the same for every reference. For example there a few ones that the journal abbreviation does not have dots.

Thank you for the reviewer's suggestion. We revised it.

Supplementary Info

Table S2

1. Please inform the reader what do the terms endogenous and efficiency are expressing. How can someone calculate them?

Thank you for the reviewer's suggestion. We changed as below.

"Endogenous (%)"  "Endogenous DNA (Unique mapped read percentage)"
"Efficiency (%)"  "Percentage except for duplications"

Text S2

1. In some case the μ l are given as ml e.g. 60 ml EB (perhaps a PDF conversion fault).

Thank you for the reviewer's comment. We changed all the mistakes in Sup Info involving Text S2.

2. In the part where there are different petrous parts used (C1, C2, C3) it is not clear which one did you used for NGS libraries, with what method and in which lab.

You mention We used [Extract2] of C1 and C3 to construct NGS libraries. And then below: Two different protocols of construction NGS libraries from five elutes of C2 ([Rinsed supernatant], [Pre-digestion], [Extract 1], [Extract 2], [Extract 3])...

This suggest this part to be written in order to be less confusing.

Thank you for the reviewer's comment. Yes, we agree that the parts were described in a confusing way. We changed the two paragraphs which start from "The petrous bone was cut....."

3. The part from "To remove large DNA fragment..." to "...for DNA solution" is very confusing. Please rephrase.

Thank you for the reviewer's comment. Yes, we removed "for DNA fragment solution."

Text S3

1. Where can someone find this customizable NGS pipeline that is mentioned? Is it described below or in another paper perhaps? Are any in-house scripts used and not given to the reader in a way that he will not be able to replicate the results? Please elaborate.

Thank you for the reviewer's comment. We added the details of NGS pipeline on the Sup Text.

2. Some references are missing e.g. BWA and others.

Thank you for the reviewer's suggestion. We put the references on the manuscript of Text S3.

3. Please ensure that in all the cases that you are mention the version of a software tool (here and in the main text) are following the same format (e.g. ver. 1.0.0 vs v1.0.0 vs -1.0.0).

We have now used consistent format for version names across the manuscript

Text S4

1. I think you should mention in every case after you merged IK002 with the different genome databases how many SNPs had in common. Are all the analyses using the same number of SNPs? If not please inform the reader when the number changes.

Thank you for the reviewer's suggestion. We added the Sup Tables including the number of SNPs.

2. Please provide a rationale to the reader to help understand why did you select the Ainu people in present-day populations and the Chokhopani 1 and Tiányuán individuals in ancient populations in ADMIXTURE analysis.

Thank you for the reviewer's suggestion. All modern and ancient individuals in the merged dataset were used in the analysis, and we changed the description accordingly."

3. The first sentence in the TreeMix paragraph is confusing. Please rephrase.

Thank you for the reviewer's suggestion. We separated the sentence into two sentences.

Text S5

1. This entire text section is not linked to the main article. Maybe somewhere in the Discussion?

Thank you for the reviewer's suggestion. But, we want to keep this in Text S5, because this information is important to interpret the genome data and too long to insert in the Discussion session. And also, we noticed that in the Discussion session of the main text, we cited Text S1 instead of Text S5. Therefore, probably the reviewer thought that Text S5 is not linked to the main article. We revised the Discussion session of the main text.

References

1. The format here is not the same with the one used in the main article.

Thank you for the reviewer's suggestion. We changed the format.

2. Baba et al. 1991: Can the journal name be abbreviated?

Thank you for the reviewer's suggestion. But, we cannot do it, because this is a Japanese journal (but, in English).

Reviewer #3 (Remarks to the Author)

The manuscript "Jomon genome sheds light on East Asian population history" by Gakuhari et al. describes production and analysis of genome-wide data from a ~2.5 kya individual from Honshu, Japan, associated with the hunter-gatherer-fisher Jomon culture that is hypothesized to represent the descendants of the earliest Upper Paleolithic settlers of the Japanese archipelago.

This represents the first Jomon genome with appreciable coverage from mainland Japan, although a low coverage Jomon genome from Honshu and two higher coverage ones from Hokkaido have been published previously. Analysis of this individual, IK002, albeit at lower coverage, has been published earlier in McColl et al. 2018.

The main results of the study are that:

1) Jomon represent an early diverging lineage basal to the split of Tibetans and the ancestors of other East Asians, Eastern Siberians and Native Americans, but dating after the split of the lineage of which Tianyuan (40kya) is a part of. As split of the ancestors of East Asians and Native Americans is dated to around 26kya, the window of divergence is consistent with Jomon being the descendants of Upper Paleolithic settlers. This is consistent with the findings of Kanzawa-Kiriyama et al. 2019.

2) The ancestors of Jomon took a southern route from South East Asia to reach the archipelago (consistent with Kanzawa-Kiriyama et al. 2019) and the Jomon shows no evidence of admixture with representatives of a northern route (here represented by the Siberian MA1). Similarities in microblade technology across northern Eurasia and in Japan in the UP could have arisen through cultural diffusion or demic diffusion by a group not related to MA1.

3) Jomon show genetic affinity to modern and ancient coastal, but not inland, populations of East Asia, possibly supporting a coastal route of spread for this early ancestry.

4) As previously described, modern Ainu are largely direct descendants of the Jomon with limited recent Japanese (East Asian-derived) gene-flow (Kanzawa-Kiriyama et al. 2019), while Japanese have a smaller fraction of Jomon ancestry (McColl et al. 2018).

The wet lab processes and computational analyses are sound and according to best practices. The argumentation is easy to follow and supported by the results. The authors do not overstate their conclusions and present alternative hypotheses where appropriate, also taking into account archaeological and anthropological evidence.

The weakness of the paper is twofold: First, limiting the analysis to only the IK002 Jomon genome while other Jomon genomes have been published. This might be due to the very recent publication of these and lack of time to redo and add analyses. For a resubmission I suggest including the additional Jomon genomes which could also address further questions of Jomon population history, e.g. overall population size, population size differences in the different islands, contact between islands and between islands and mainland, dating/location of gene-flow into modern Japanese etc.

This would help with the second weakness, the large overlap with recent publications. Additional analyses could set the paper apart from previously published ones, especially McColl et al. 2018 and Kanzawa-Kiriyama et al. 2019, as many of the findings here are not exactly novel or surprising.

Furthermore, the authors could address and explain differences in their findings to the previous papers, such as different admixture proportions of Jomon into Ainu and modern Japanese, or the finding in McColl that "the Jōmon individual is best-modeled as a mix between a population related to group 1/Önge and a population related to East Asians" rather than a deep diverging lineage.

- Concerning the Jomon-like ancestry in Ulchi and Nivkh, to distinguish the scenarios "This could be an influence of the Hokkaido Ainu who are likely to be direct descendants of the Jomon people. The ancestor of the Ainu people could have admixed with the Okhotsk people⁵⁵ who were genetically as well as morphologically close to Ulchi and Nivkh in the the Primorye region^{56–60}. The second explanation is that the track of the earliest-wave was erased in the inland but left over in the coastal region.", can you use ALDER to date the admixture event? Same for the Jomon-like ancestry in Devil's Gate. Are these events independent?

Thank you for the reviewer's suggestion. We have now included an ALDER analysis using Devil's Gate Cave and Jomon as source groups for dating the admixture in the Ulchi, and find evidence for recent admixture date of 26 generations ago, thereby supporting the first scenario. We also ran ALDER on Devil's Gate cave as a target, but find no significant LD curve. Together this suggests that Jomon-like ancestry in Devil's Gate cave is due to older admixture Jomon-related peoples, and hence different from the recent admixture observed in Ulchi.

Suggested edits and small mistakes that should be fixed:

-mention the geographic origin of the sample in the abstract

Thank you for the reviewer's suggestion. We changed as below.

Here, we analyze the whole-genome sequence of a 2.5 kya individual (IK002) characterized with a typical Jomon culture.

Here, we analyze the whole-genome sequence of a 2.5 kya individual (IK002) from the main island of Japanese archipelago characterized with a typical Jomon culture.

-Some of the results are referenced for the first time in the discussion, such as the qpgraph modeling. These should get their own paragraph in the results.

Thank you for the reviewer's suggestion. We have moved this paragraph to the results section

-Fig. 1: The labeling in B is hardly legible; Loschbour is a Mesolithic individual from West Eurasia not Central Siberia

Thank you for the reviewer's comments. We have updated the figure accordingly

-Fig. 4 needs more explanation in the caption. What is bEE, a population that admixed with the ancestors of IK002, East Asians and NEAsians/East Siberians, or an encompassing term to describe the basal lineage from which the lineages leading to IK002, East Asians and NEAsians/East Siberians branched off?

Thank you for the reviewer's comments. In the figure legends, we changed as below.

"The basal East Eurasians (bEE) represent an ancient population that had no divergence among the ancestors of East Asians, Northeast Asians/East Siberian, and Native Americans. NA-ES-NA presents another ancient

population that had no split between the ancestors of Northeast Asians/East Siberian and Native Americans."

-Fig. S4: Add a separate legend. Jomon samples are called F23 and F5 in caption, but fun23 and fun5 in plot

We have removed this figure from the supplement as the Hokkaido Jomon samples are now included in the main Fig. 1

-Fig. S7: duplicated sentence in caption: "The scale bar shows the average standard error (SE) of the entries in the covariance matrix."

Thank you for the reviewer's comment. We removed one of them.

-Fig. S8: A and B switched in caption; Ust'Belaya mapped in wrong location, should be near Lake Baikal

Thank you for the reviewer's comment. We switched ancient and modern in the caption, and updated the plot with the correct location for Ust'Belaya

-The texts especially of the supplementary material could use a round of copyediting, there are numerous typos or grammatical mistakes, e.g. in Materials and Methods: "we chose IK002, who were excavated from" ("were" > "was"). The main text looks mostly fine.

Thank you for the reviewer's comment. We revised the mistake.

-As a visual summary of the results, a schematic such as Fig. 4 in McColl et al. 2018 (but focusing on the settlement waves of Japan and dispersal of Jomon-like ancestry) would be a nice addition, especially since this study deals with routes of dispersal.

Thank you for the reviewer's comment. But, instead of figure like Fig.4 in McColl et al. 2018, we show Fig.4 in our manuscript, because we could see the divergence timing (tree topology) but find no route that the Jomon ancestry went north. This would be published in the next paper that includes more individuals from the archipelago.

Reviewers' comments:

Reviewer #1 (Remarks to the Author):

The authors have substantially revised and improved the manuscript with additional supporting analysis. However, there are important concerns that needs to be addressed:

1) For comment item number 5, it wasn't addressed how consistent the iterations are for each K or how the 'best runs' were chosen. Does K=15 have the best support? Did it produce 10/10 or consistent results in all 10 runs? It is best to place the proportion of support for each K (either from 1/10 to 10/10 runs) on the left side of the admixture plot. In addition, the population labels are difficult to read due to its low resolution.

2) For related comment item numbers 10 and 15. One of major claims in the manuscript is that Jomon is basal to all East & Northeast Asians. However, an alternative model that can explain otherwise was not tested, where Jomon is an admixture between Onge/Hoabinhian-related and Ami-related ancestries. This can easily be tested and be compared with the admixture graph models in Fig S11. The base model in Fig. S10 can be misleading, as the Ami population was not included in the topology, which is very important given the f4 results where there is evident gene flow between Jomon and Ami (or any other least admixed East Asians). They can construct an alternative model to the leftmost graph of fig S11, where Jomon is placed as an admixture between Onge and Ami (instead of the other way around), and assess which has a better statistical support. They can also test $f_4(\text{Mbuti}; \text{Onge}, \text{Japanese}/\text{Han}/\text{etc}, \text{Jomon})$ to determine whether there is excess allele sharing between Onge and Jomon relative to present-day Japanese/Han/etc. The shared ancestry component can be seen as the "grey cluster" found in both Jomon and Lao-Hoabinhian from K5-K10 of the admixture plot. This is especially important given the publication by McColl et al. and the recent release of the preprint by Wan et al., that consistently show Jomon as an admixture between Taiwanese aborigine-related and deeply diverging Onge-related ancestries, which disputes the core claim of this paper.

<https://www.biorxiv.org/content/10.1101/2020.03.25.004606v1>

Reviewer #2 (Remarks to the Author):

All my concerns about the first submission have been satisfactory addressed. Thus, I find the revised manuscript fitted to be published in Communications Biology.

Answer to the reviewer#1

1) For comment item number 5, it wasn't addressed how consistent the iterations are for each K or how the 'best runs' were chosen. Does K=15 have the best support? Did it produce 10/10 or consistent results in all 10 runs? It is best to place the proportion of support for each K (either from 1/10 to 10/10 runs) on the left side of the admixture plot. In addition, the population labels are difficult to read due to its low resolution.

Answer:

We have clarified the methods description on how the best runs were chosen, it now reads

“ADMIXTURE was run in 10 replicates, for K values ranging from K=2 and K=20. Major modes across replicates were determined and aligned across different K values using pong.”

We also indicated the number of replicates consistent with the major mode for each K in the figure as requested.

K Replicates

2	10/10
3	10/10
4	10/10
5	10/10
6	9/10
7	5/10
8	10/10
9	4/10
10	3/10
11	5/10
12	9/10
13	6/10
14	6/10
15	5/10
16	4/10
17	5/10
18	5/10

19 4/10

20 4/10

2) For related comment item numbers 10 and 15. One of major claims in the manuscript is that Jomon is basal to all East & Northeast Asians. However, an alternative model that can explain otherwise was not tested, where Jomon is an admixture between Onge/Hoabinhian-related and Ami-related ancestries. This can easily be tested and be compared with these admixture graph models in Fig S11. The base model in Fig. S10 can be misleading, as the Ami population was not included in the topology, which is very important given the f4 results where there is evident gene flow between Jomon and Ami (or any other least admixed East Asians). They can construct an alternative model to the leftmost graph of fig S11, where Jomon is placed as an admixture between Onge and Ami (instead of the other way around), and assess which has a better statistical support. They can also test $f_4(\text{Mbuti}; \text{Onge}, \text{Japanese/Han/etc}, \text{Jomon})$ to determine whether there is excess allele sharing between Onge and Jomon relative to present-day Japanese/Han/etc. The shared ancestry component can be seen as the “grey cluster” found in both Jomon and Lao-Haobinhian from K5-K10 of the admixture plot. This is especially important given the publication by McColl et al. and the recent release of the preprint by Wan et al., that consistently show Jomon as an admixture between Taiwanese aborigine-related and deeply diverging Onge-related ancestries, which disputes the core claim of this paper.

Answer:

We thank the reviewer’s comment. We have now added qpGraph analyses for the alternative model as a new supplementary figure (Fig. S12). The new analyses show that both alternatives have equivalent support in qpGraph, as there is a lack of additional samples in that part of the graph to further constrain the topologies. We nevertheless point out the following observations:

- In either model the ancestry of Jomon is deep and an outgroup to other East and NE Asians (Chokhopani, Shamanka), consistent with our main conclusion.
- The length of the edge for shared drift between the Onge and the Jomon-contributing ancestry in the Jomon admixed model (i.e. McColl et al.

2018) is 0, so there is no apparent affinity to Onge. This is also confirmed by direct f_4 tests as suggested by the reviewer, which are included in a new supplementary table S9.

- While the recent preprint of Wang et al also models Jomon as admixed between a deep Onge-like lineage, their admixture graph models the non-Onge ancestry as derived from Amur basin populations (Boisman/Devil's Gate cave), which in our model is admixed from a Jomon-related source. Independently of which of the models is employed, the conclusion reached are remarkably similar, again supporting Jomon to derived from a deep East Asian lineage with contributions to other coastal populations.

We have therefore added the following discussion on these observations to the final paragraph of Results section in the manuscript:

“We note that we fit the backbone graph assuming an unadmixed Jomon lineage, as opposed to a previously suggested dual-ancestry model where Jomon is admixed between Önge- and Ami-related ancestry. This alternative base model provides an equivalent admixture graph fit, however we find no evidence for shared genetic drift between the Önge and the ancestral Jomon lineage in qpGraph (Fig.S12a & Fig.S12b), or using direct f_4 statistics (Table S9 & Table S10). Additional sampling of early East Asian human remains will be needed to further resolve the relationships among these deep lineages, but nevertheless either model supports the deep origins of Jomon.”

REVIEWERS' COMMENTS:

Reviewer #1 (Remarks to the Author):

The revised manuscript has adequately address the recent concerns.